# Unsupervised learning of disentangled multiscale representations via score-based variational autoencoders

## Abstract

We present the Score-based autoencoder for Multiscale Inference (SAMI), a method for unsupervised representation learning that combines the theoretical frameworks of diffusion models and VAEs. By unifying their respective evidence lower bounds, SAMI formulates a principled objective that learns representations through score-based guidance of the underlying diffusion process. The resulting representations automatically capture meaningful structure in the data: it recovers ground truth generative factors in our synthetic dataset, learns factorized, semantic latent dimensions from complex natural images, and encodes video sequences into latent trajectories that are straighter than those of alternative encoders, despite training exclusively on static images. Furthermore, SAMI can extract useful representations from pre-trained diffusion models with minimal additional training. Finally, the explicitly probabilistic formulation provides new ways to identify semantically meaningful axes in the absence of supervised labels, and its mathematical exactness allows us to make formal statements about the nature of the learned representation. Overall, these results indicate that implicit structural information in diffusion models can be made explicit and interpretable through synergistic combination with a variational autoencoder.

## 1 Introduction

To evaluate the behavioral relevance of their observations intelligent agents must possess a notion of semantics. This requires decomposing complex sensory observations into abstract and semantically meaningful factors of variation, a capability known in machine learning as disentangling (Bengio et al., 2014). Disentangled representations promise several advantages, including improved interpretability, enhanced generalization, and stronger transfer capabilities across related tasks (Higgins et al., 2017). However, given that for both biological and artificial agents, the amount of unlabeled data vastly exceeds that of labeled data, an important consideration is how to learn disentangled representations without relying on explicit supervision.

Latent variational models provide a principled framework grounded in probabilistic generative models. The variational auto-encoder (VAE) has emerged as a particularly promising instantiation of this idea (Kingma and Welling, 2013; Rezende et al., 2014), which allows for amortized inference of latent structure and data generation through optimization of the evidence lower bound (ELBO). Successful variants of this approach have been used for compression (Ballé et al., 2017; Ballé et al., 2021), prediction (Tishby et al., 2000; Alemi et al., 2019; Sachdeva et al., 2020), and as models of neural activity in visual cortex (Csikor et al., 2022; Vafaii et al., 2023). Crucially, VAEs can be encouraged to exhibit disentangled representations via appropriate regularization, as with $\beta$-VAEs (Higgins et al., 2017; Alemi et al., 2019).

Despite these successes, there is a fundamental tension between reconstruction fidelity and disentanglement quality (Kumar et al., 2018; Sikka et al., 2019) in such models. Encouraging disentangled representations typically requires stronger regularization that limits the expressiveness of latent variables; this trade-off between compression and reconstruction precision is well known as the rate-distortion principle. This limitation has motivated various VAE improvements, including enhanced posterior approximations through non-diagonal covariances or complex posterior distributions

(Manduchi et al., 2023; Klushyn et al., 2019; Mathieu et al., 2019; Cheng et al., 2020; Rezende and Mohamed, 2015), additional latent constraints (Chen et al., 2019; Zhao et al., 2018), and more flexible latent priors (Klushyn et al., 2019; Wehenkel and Louppe, 2021). Notably, these efforts have largely focused on the inference mechanism rather than the generative model itself, despite the role the diagonal posterior covariance plays in encouraging disentanglement (Rolínek et al., 2019; Dai et al., 2018) and the importance of the generative model in determining the nature of learned representations (Cremer et al., 2018). Moreover, mathematical analyses of VAE optimality typically assume that the data lies on a manifold with fixed intrinsic dimensionality Dai and Wipf (2018), but such an assumption is problematic for natural images, which instead appear to lie in a multi-scale manifold whose intrinsic dimensionality is highly dependent on the signal to noise ratio (SNR) of the image (Guth et al., 2025; Sclocchi et al., 2025). This suggests that to learn good disentangled representations of complex, multi-scale data, we need generative models with the appropriate inductive biases Locatello et al. (2019).

A possible avenue for building more expressive, multi-scale generative models lies in diffusion models (Sohl-Dickstein et al., 2015; Ho et al., 2020). These generative models produce high-quality samples in many domains by learning to approximate the data distribution through a denoising objective, and seemingly captures features at multiple spatial scales (Sclocchi et al., 2025). However, their lack of an explicit latent representation poses challenges for tasks requiring embeddings, such as disentangled representation learning. Additionally, guiding the diffusion sampling process to generate samples with specific attributes or classes remains an open research question (Fuest et al., 2024). While techniques like classifier-guided diffusion (Dhariwal and Nichol, 2021) and classifier-free guidance (Ho and Salimans, 2021) lay the groundwork for sampling from conditional densities in the presence of explicit class labels, achieving robust and flexible conditioning without compromising sample quality or diversity is not fully resolved (Chidambaram et al., 2024; Sadat et al., 2024; Kaiser et al., 2024; Ifriqi et al., 2025). Critically, few studies explore the learning of latents that enable diffusion models to effectively navigate the full data manifold *without auxiliary information*.

Here, we introduce a Score-based Autoencoder for Multiscale Inference (SAMI) that combines the strengths of variational autoencoders and diffusion models to achieve unsupervised learning of structured, interpretable latent representations while maintaining high-quality sample generation. Our key innovation lies in employing conditional diffusion as the generative component of a VAE, coupled with a precise mathematical formalization of the objective via the ELBO that enables reuse of the inference network for conditioning.

Leveraging the mathematical exactness of our model, we prove that the diffusion prior on the data space allows us to keep a simple latent posterior approximation that imposes strong factorization constraints on the latent code such that multi-scale generative features are disentangled. We also prove that learning latent representations of images that are coherent across noise levels helps to unify and smooth the latent representation.

We find empirical support for these results. When trained on image data, our framework yields semantically meaningful latent axes and produces straighter trajectories when encoding natural video sequences, all without compromising generative performance. Moreover, a distinctive feature of our approach is that the posterior structure enables identification of semantically meaningful axes without reliance on supervised labels. From the diffusion perspective, incorporating an inference network with an explicit representational space facilitates unsupervised discovery of factorized and interpretable latents that can be used to control sample generation. Notably, this same methodology can be applied to extract semantic latent representations from pretrained diffusion models.

## 2 METHODS

The task of extracting latent representations from data can be formalized as a graphical model in which latent random variables, $\mathbf{z} \in Z$, give rise to observations $\mathbf{x} \in X$ (Fig. 1A, yellow box). Representational learning then corresponds to maximizing the marginal likelihood of the observed data under the model, $\prod_k \int p_\theta(\mathbf{x}_k, \mathbf{z}) d\mathbf{z}$, while inference requires calculating the posterior distribution $p(\mathbf{z}|\mathbf{x})$ via Bayes' rule. Unfortunately, directly computing the log likelihood and the posterior distribution in closed form is usually infeasible for expressive models.

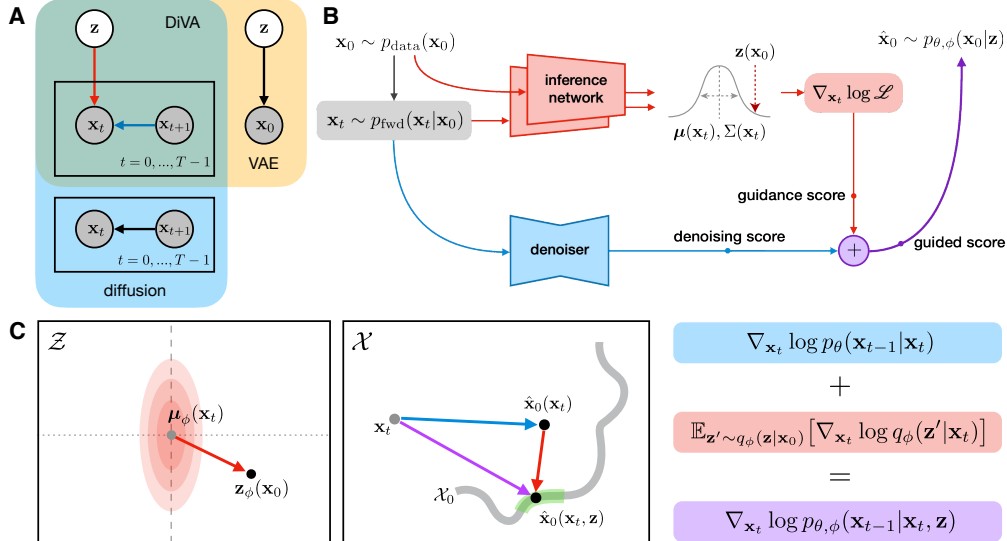

Figure 1: **A)** Graphical model for our diffusion VAE (SAMI), contrasted with those of standard VAEs and unconditioned diffusion models. **B)** Schematic of conditional sampling procedure. **C)** Schematic illustration of how movement in the latent space results in guidance of the denoiser score towards regions of the clean image manifold that are semantically similar to the original image. For visualization, both latent and image spaces are depicted as two dimensional.

Variational autoencoders (VAEs) address this difficulty by approximating the posterior with a simpler parametric form, whose parameters are data-specific and obtained via optimization (Kingma and Welling, 2013; Rezende et al., 2014). The parameters of both the amortized posterior and the generative model are learned via minimization of the *evidence lower bound* (ELBO), a tractable lower bound on the model marginal likelihood:

$$\mathcal{L}_{\text{ELBO}}(\theta, \phi; \mathbf{x}) = \mathbb{E}_{\mathbf{z} \sim q_\phi(\mathbf{z}|\mathbf{x})} \left[ \log \frac{p_\theta(\mathbf{x}, \mathbf{z})}{q_\phi(\mathbf{z}|\mathbf{x})} \right] \leq \log p_\theta(\mathbf{x}), \tag{1}$$

where $q_\phi(\mathbf{z}|\mathbf{x})$ is the approximate posterior (usually computed by a neural network with weights $\phi$), the density $p_\theta(\mathbf{x}, \mathbf{z}) = p_\theta(\mathbf{x}|\mathbf{z})p(\mathbf{z})$ is the generative model, with prior $p(\mathbf{z})$ given by a simple parametric distribution (e.g., a standard Gaussian), and likelihood $p_\theta(\mathbf{x}|\mathbf{z})$ computed by a separate neural network with weights $\theta$. In practice, the limitations imposed on the likelihood by this parameterization often lead to poor quality of generated samples (Burda et al., 2016; Sønderby et al., 2016; Rezende and Viola, 2018).

In contrast, diffusion models are highly expressive generative models that can estimate and sample from complex densities. Diffusion models such as denoising diffusion probabilistic models (DDPMs) form an implicit prior over data by learning to denoise noisy versions of clean data (Sohl-Dickstein et al., 2015; Kadkhodaie and Simoncelli, 2020; Ho et al., 2020). Noisy data are generated by a fixed *forward* operator that is assumed to be additive Gaussian, $\mathbf{x}_t \sim p_{\text{fwd}}(\mathbf{x}_t|\mathbf{x}_0)$, with $t \in [0, 1, ..., T]$ specifying the noise variance. This denoising objective can be written as the mean squared error between the model's estimate of the noise $\boldsymbol{\epsilon}_\theta(\mathbf{x}_t, t)$ and the true noise present in the noisy image $\boldsymbol{\epsilon}$, which is typically assumed to be isotropic Gaussian. Minimizing this error has also been shown to be equivalent to maximizing the ELBO (up to a noise-dependent scalar weighting $\lambda_t$) for a joint data distribution given by $p(\mathbf{x}_{0:T})$ and a joint "posterior" $q(\mathbf{x}_{1:T}|\mathbf{x}_0)$ that captures the distribution over all noising paths (Kingma and Gao, 2023):

$$\mathcal{L}_{\text{DDPM}} = \mathbb{E}_{\boldsymbol{\epsilon} \sim \mathcal{N}(0, I), \, t \sim [0, T]} \left[ \lambda_t \|\boldsymbol{\epsilon} - \boldsymbol{\epsilon}_\theta(\mathbf{x}_t, t)\|^2 \right] = -\mathbb{E}_{q(\mathbf{x}_{1:T}|\mathbf{x}_0)} \left[ \log \frac{p_\theta(\mathbf{x}_{0:T})}{q(\mathbf{x}_{1:T}|\mathbf{x}_0)} \right]. \tag{2}$$

It is well known that the minimum mean squared error (MMSE) estimate is given by the posterior mean, and this in turn can be related to the score function via Miyasawa's/Tweedie's formula (Robbins, 1956; Miyasawa et al., 1961). Under a Gaussian variance-preserving noise distribution given by $p(\mathbf{x}_t|\mathbf{x}_0) = \mathcal{N}(\mathbf{x}_t|\sqrt{\bar{\alpha}_t}\mathbf{x}_0, (1-\bar{\alpha}_t)I)$, where $\{\bar{\alpha}_t\}$ defines the variance schedule, the MMSE estimate

of the noise is given as

$$\hat{\boldsymbol{\epsilon}}_\theta(\mathbf{x}_t) = \underset{\boldsymbol{\epsilon}_\theta}{\operatorname{argmin}} \|\boldsymbol{\epsilon} - \boldsymbol{\epsilon}_\theta(\mathbf{x}_t, t)\|^2 = \mathbb{E}_{p(\mathbf{x}_0|\mathbf{x}_t)}[\boldsymbol{\epsilon}] = -\sqrt{1 - \bar{\alpha}_t} \nabla_{\mathbf{x}_t} \log p(\mathbf{x}_t). \tag{3}$$

Once trained, generation of samples relies on an iterative partial denoising process that samples $\mathbf{x}_t \sim p_\theta(\mathbf{x}_t|\mathbf{x}_{t+1})$ as $t = T - 1, ..., 0$ (Fig. 1A, blue box). Examples from the training set can be thought of as samples from a complex probability distribution that resembles a low-dimensional, nonlinear manifold lying within the high-dimensional data space (Ho et al., 2020; Kadkhodaie and Simoncelli, 2020), and the iterative denoising process effectively learns to project noisy samples back onto this manifold. To generate samples with desired characteristics, the denoiser can be guided towards specific regions of the manifold through classifier-based or embedding-based guidance signals (Dhariwal and Nichol, 2021). However, unlike variational autoencoders or GANs (Goodfellow et al., 2014), diffusion models do not specify or infer an explicit latent representation.

By using conditional diffusion models as part of the VAE generative process, we can overcome the limitations of both models. The key idea is to learn the latent variables that will best guide the diffusion process towards the observations. Concretely, we learn a latent representation by augmenting the Markov chain $\mathbf{x}_{0:T}$ of the diffusion model with a latent variable $\mathbf{z}$ that guides the denoising process (Fig. 1A, green box). This results in a joint distribution $p(\mathbf{x}_{0:T}, \mathbf{z})$ and approximate posterior $q(\mathbf{x}_{1:T}, \mathbf{z}|\mathbf{x}_0)$ that can be factorized as

$$p(\mathbf{x}_{0:T}, \mathbf{z}) = p_{\theta,\phi}(\mathbf{x}_{0:T}|\mathbf{z})p(\mathbf{z})$$
$$q(\mathbf{x}_{1:T}, \mathbf{z}|\mathbf{x}_0) = q_{\mathbf{x}}(\mathbf{x}_{1:T}|\mathbf{x}_0)q_\phi(\mathbf{z}|\mathbf{x}_0),$$

where the latter holds because the noisy samples in $\mathbf{x}_{1:T}$ are independent of $\mathbf{z}$ when conditioned on $\mathbf{x}_0$. Here, $q_\phi(\mathbf{z}|\mathbf{x}_0)$ is the inference network, used to infer latent $\mathbf{z}$ from observations $\mathbf{x}_0$, while $q_{\mathbf{x}}(\mathbf{x}_{1:T}|\mathbf{x}_0)$ is the (known) "posterior" that defines the conditional forward process of the diffusion model. As is common for VAEs, we assume that the model prior is an isotropic Gaussian, $p(\mathbf{z}) = \mathcal{N}(\mathbf{0}, \mathbf{I})$. Incorporating these into the ELBO yields a joint objective function of the form:

$$\mathcal{L}_{\text{SAMI}} = -\mathbb{E}_{q_\phi(\mathbf{z}|\mathbf{x}_0)}\mathbb{E}_{q_{\mathbf{x}}(\mathbf{x}_{1:T}|\mathbf{x}_0,\mathbf{z})}\left[\log \frac{p_{\theta,\phi}(\mathbf{x}_{0:T}|\mathbf{z})p(\mathbf{z})}{q_{\mathbf{x}}(\mathbf{x}_{1:T}|\mathbf{x}_0)q_\phi(\mathbf{z}|\mathbf{x}_0)}\right]$$

$$= -\mathbb{E}_{q_\phi(\mathbf{z}|\mathbf{x}_0)}\mathbb{E}_{q_{\mathbf{x}}(\mathbf{x}_{1:T}|\mathbf{x}_0,\mathbf{z})}\left[\log \frac{p_{\theta,\phi}(\mathbf{x}_{0:T}|\mathbf{z})}{q_{\mathbf{x}}(\mathbf{x}_{1:T}|\mathbf{x}_0)}\right] - \mathbb{E}_{q_\phi(\mathbf{z}|\mathbf{x}_0)}\left[\log \frac{p(\mathbf{z})}{q_\phi(\mathbf{z}|\mathbf{x}_0)}\right]$$

$$= \mathbb{E}_{q_\phi(\mathbf{z}|\mathbf{x}_0)}\left[D_{\text{KL}}\left(q_{\mathbf{x}}(\mathbf{x}_{1:T}|\mathbf{x}_0)\|p_{\theta,\phi}(\mathbf{x}_{0:T}|\mathbf{z})\right)\right] + D_{\text{KL}}\left(q_\phi(\mathbf{z}|\mathbf{x}_0)\|p(\mathbf{z})\right), \tag{4}$$

where the second term is equivalent to the standard regularization term in the VAE objective that encourages the expected approximate posterior distribution to be close to the latent prior. Inputs $\mathbf{x}_0$ to the inference network return the corresponding mean and covariance $\boldsymbol{\mu}_\phi(\mathbf{x}_0)$ and $\Sigma_\phi(\mathbf{x}_0)$ of the posterior distribution $q_\phi(\mathbf{z}|\mathbf{x}_0)$, which are then used to analytically calculate the KL divergence term.

**Implementing the generative model as conditional diffusion.** The first term in SAMI's objective (Eq. 4) encourages the latent conditioned generative model $p_{\theta,\phi}(\mathbf{x}_{0:T}|\mathbf{z})$ to be close to the distribution over forward noising paths $q_{\mathbf{x}}$. Following a similar derivation to Ho et al. (2020), this term can be written as the expected mean squared error between the true and estimated noise across noise levels $t$:

$$\mathcal{L}_{\text{SAMI}} = \mathbb{E}_{q_\phi(\mathbf{z}|\mathbf{x}_0),\boldsymbol{\epsilon},t}\left[\lambda_t\|\boldsymbol{\epsilon} - \boldsymbol{\epsilon}_{\theta,\phi}(\mathbf{x}_t, t, \mathbf{z})\|^2\right] + D_{\text{KL}}\left(q_\phi(\mathbf{z}|\mathbf{x}_0)\|p(\mathbf{z})\right), \tag{5}$$

where $\boldsymbol{\epsilon}$ is the true noise, $\boldsymbol{\epsilon}_{\theta,\phi}$ is the function estimating it, and $\lambda_t$ is a scalar hyperparameter that depends on $t$. For the first term in the objective, we can use Miyasawa's formula to relate the conditional MMSE estimate to the guidance score (full derivation in Appendix A.1). Using the same variance-preserving noise distribution as in Eq. (3), we see that

$$\hat{\boldsymbol{\epsilon}}_{\theta,\phi}(\mathbf{x}_t, \mathbf{z}) = \underset{\boldsymbol{\epsilon}_{\theta,\phi}}{\operatorname{argmin}} \|\boldsymbol{\epsilon}_t - \boldsymbol{\epsilon}_{\theta,\phi}(\mathbf{x}_t, t, \mathbf{z})\|^2 = -\sqrt{1 - \bar{\alpha}_t}\, \nabla_{\mathbf{x}_t} \log p(\mathbf{x}_t|\mathbf{z}). \tag{6}$$

Critically, Bayes' rule can be used to express the guidance score as the sum of two terms, an unconditional denoiser score and a guidance score:

$$\nabla_{\mathbf{x}_t} \log p(\mathbf{x}_t|\mathbf{z}) = \nabla_{\mathbf{x}_t} \log p(\mathbf{x}_t) + \nabla_{\mathbf{x}_t} \log p(\mathbf{z}|\mathbf{x}_t). \tag{7}$$

Plugging this back into Eq. (6), we see that the MMSE solution can be expressed as

$$\hat{\boldsymbol{\epsilon}}_{\theta,\phi}(\mathbf{x}_t, \mathbf{z}) = -\sqrt{1 - \bar{\alpha}_t}\left(\nabla_{\mathbf{x}_t} \log p(\mathbf{x}_t) + \nabla_{\mathbf{x}_t} \log p(\mathbf{z}|\mathbf{x}_t)\right)$$

$$= \hat{\boldsymbol{\epsilon}}_\theta(\mathbf{x}_t) - \gamma_t\, \mathbf{g}_t(\mathbf{x}_t, \mathbf{z}), \tag{8}$$

where $\hat{\epsilon}_\theta(\mathbf{x}_t) = -\sqrt{1-\bar{\alpha}_t}\,\nabla_{\mathbf{x}_t}\log p(\mathbf{x}_t)$ is the MMSE solution associated with an unconditional diffusion model (Eq. 3), $\mathbf{g}_t(\mathbf{x}_t, \mathbf{z}) = \nabla_{\mathbf{x}_t}\log p(\mathbf{z}|\mathbf{x}_t)$ is the guidance score, and $\gamma_t = \sqrt{1-\bar{\alpha}_t}$ is a noise level-dependent scalar term that weights the two terms appropriately.

**Computing the guidance score.** From the perspective of the unconditional diffusion model, the VAE posterior has a dual interpretation as a likelihood function $\mathscr{L}_\phi(\mathbf{x}_t; \mathbf{z})$, which is a function of the noisy image $\mathbf{x}_t$ over the latent domain $\mathcal{Z}$. The guidance score is the score of the likelihood evaluated at a particular latent value $Z = \mathbf{z}$, which in our case we take to be the latent corresponding to the clean image, since our goal is to reconstruct the original observation. To compute this, we take samples of the clean image latent $\mathbf{z}(\mathbf{x}_0) \sim q_\phi(\mathbf{z}|\mathbf{x}_0)$ in accordance with the expectation in Eq. (5) and compute its log likelihood, $\log \mathscr{L}_\phi(X_t = \mathbf{x}_t; Z = \mathbf{z}(\mathbf{x}_0))$. Taking the derivative of this scalar quantity with respect to the noisy image $\mathbf{x}_t$ (via auto-diff) then gives the approximate guidance score

$$\mathbf{g}_{t,\phi}(\mathbf{x}_t, \mathbf{z}) = \mathbb{E}_{\mathbf{z}' \sim q_\phi(\mathbf{z}|\mathbf{x}_0)}\big[\nabla_{\mathbf{x}_t}\log q_\phi(\mathbf{z}'|\mathbf{x}_t)\big] \tag{9}$$

The reuse of the inference network to compute the log likelihood removes the need for a separate neural network to map the representation $\mathbf{z}$ into a guiding signal, and ties the representations of clean and noisy versions of an image together. Plugging this back into Eq.(8), and this in turn into Eq. (5), we see that the SAMI objective can be written as the influence of two separate networks in our model, the unconditional denoiser parameterized by $\theta$, and the inference network parameterized by $\phi$:

$$\mathcal{L}_{\beta\text{-SAMI}} = \mathbb{E}_{q_\phi(\mathbf{z}|\mathbf{x}_0), \boldsymbol{\epsilon}, t}\Big[\lambda_t \|\boldsymbol{\epsilon}_t - \boldsymbol{\epsilon}_\theta(\mathbf{x}_t, t) + \gamma_t\,\mathbf{g}_{t,\phi}(\mathbf{x}_t, \mathbf{z})\|^2\Big] + \beta\,D_{\mathrm{KL}}\big(q_\phi(\mathbf{z}|\mathbf{x}_0)\|p(\mathbf{z})\big), \tag{10}$$

where $\beta$ is a hyperparameter that balances the effective contribution of the KL regularization on the latent space. This is akin to the Lagrange multiplier that controls the reconstruction and regularization terms in $\beta$-VAEs (Higgins et al., 2017). Note that if we rewrite this derivation in terms of $\mathbf{x}_0$ estimates instead of $\boldsymbol{\epsilon}$ estimates, we can show, using the relation $\mathbf{x}_t = \sqrt{\bar{\alpha}_t}\,\mathbf{x}_0 + \sqrt{1-\bar{\alpha}_t}\,\boldsymbol{\epsilon}$, that $\hat{\mathbf{x}}_0(\mathbf{x}_t, \mathbf{z}) = \hat{\mathbf{x}}_0(\mathbf{x}_t) + {1-\bar{\alpha}_t}/{\sqrt{\bar{\alpha}_t}}\mathbf{g}_t(\mathbf{x}_t, \mathbf{z})$, which reflects the additive relation between the denoising score and guidance score in Fig. 1C, $\mathcal{X}$ space. The expression that details the effect of the guidance score on the DDPM reverse transition operator is given in Appendix A.2.

**Inference and guidance.** Once SAMI is trained using Eq. (10), inference follows the general VAE prescription where latent samples are drawn from the posterior distribution $q_\phi(\mathbf{z}|\mathbf{x}_0)$ via the reparameterization trick. Generating image samples that resemble the given "guidance" datapoint $\mathbf{x}_0$ involves conditioning the diffusion process on draws from the posterior distribution. As in DDPMs, we start the conditional generation process with a sample from an isotropic gaussian $\mathbf{x}_t \sim \mathcal{N}(0, I)$. At each noise level $t$, we estimate the noise present in the noisy image via the denoiser $\boldsymbol{\epsilon}_\theta(\mathbf{x}_t, t)$, and compute the guidance score $\mathbf{g}_{t,\phi}$ using Eq. (9). Note that while the clean image posterior remains the same at every noise level, the guidance score changes because we evaluate the clean image latent under the noisy image posterior, which changes with the noise level. The resulting guidance score biases the reverse process according to Eq. (8), such that as the noise level goes to 0, we arrive at a sample from the conditional distribution $\hat{\mathbf{x}}_0 \sim p_{\theta,\phi}(\mathbf{x}_0|\mathbf{z})$. Algorithms are provided in Appendix A.3.

**Geometric intuition.** In the image space, conditional generation update steps (purple arrow in Fig. 1C, $\mathcal{X}$ space) can be thought of as a combination of two forces: starting from a noisy image $\mathbf{x}_t$, attractor dynamics push the state towards the image manifold (blue arrow) while the latent-based guidance selects specific regions of the manifold (red arrow). The addition of these two forces direct the network towards regions of the data manifold that are close to the target $\mathbf{x}_0$ (green shade).

In the latent space (Fig. 1C, $\mathcal{Z}$), similarity between the current state $\mathbf{x}_t$ and the conditioning image $\mathbf{x}_0$ (as defined by the likelihood) is the Mahalanobis distance between the mean estimate of the network state in latent space $\boldsymbol{\mu}_\phi(\mathbf{x}_t)$ and the latent corresponding to the conditioning image $\mathbf{z}'(\mathbf{x}_0)$, computed under a metric that is given by the inverse of the covariance $\Sigma_\phi^{-1}$ (concentric ellipsoids). Conditional generation can thus be cast as a constrained optimization process, where the network minimizes the latent Mahalanobis distance subject to the constraints posed by the geometry of the latent space and the image manifold (for further details, see Appendix A.4).

During the learning process, the network learns to map noisy version of the same image close together in $\mathcal{Z}$ space, while accounting for uncertainty introduced by the noise in the associated posterior covariance, $\Sigma_\phi(\mathbf{x}_t)$. This effectively means learning which directions in latent space correspond to noise-induced variations versus meaningful semantic content.

## 3 RESULTS

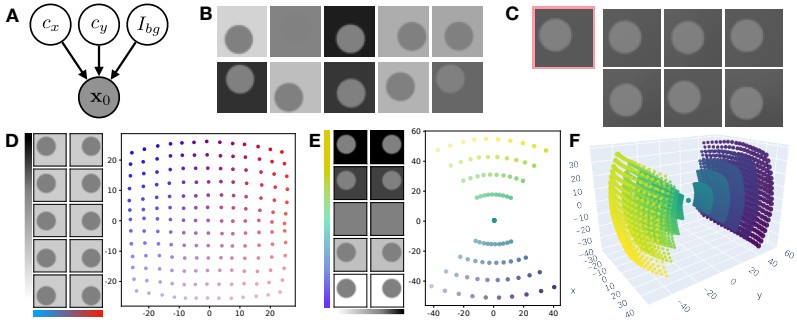

Figure 2: **Disks dataset. A)** Graphical model for the disks dataset, with coordinates of disk center $(c_x, c_y)$ and background intensity $I_b$ as latents. **B)** Random draws from the ground truth generative process. **C)** Samples drawn from the model, conditioned on leftmost image. **D)** Posterior means for a grid of test $c_x$ and $c_y$ ground truth positions, fixed $I_b$. **E)** Same as D, but $c_y$ fixed during interpolation of the other two factors. **F)** Same as D, for all three factors.

**Synthetic disks dataset.** We first investigated whether SAMI is able to learn a semantically meaningful latent space by training the network on a synthetic dataset where the ground truth latent factors and generative structure are known, and the number of latent dimensions matches that of the latent factors. The dataset is a simplified version of that used by Kadkhodaie et al. (2024), comprised of images of circles ("disks") of a fixed radius and intensity on a blank background (Fig. 2A, B). The images are fully determined by three independent generative factors: the coordinates of the disk center, $c_x, c_y$, and the intensity of the background, $I_{bg}$. All three generative factors are uniformly distributed, and the foreground intensity of the disks is held to a constant value of $0.5$.

As a first measure of whether SAMI learns a useful representation, we tested the model's ability to faithfully reconstruct images when conditioned on a given guidance image (Fig. 2C; see Appendix A.5 for architecture details). The conditioned image generation is sensible: it preserves background intensity (which can be accurately estimated from the conditioning image), with small variations in the location of the disc reflecting some posterior uncertainty in that dimension. To quantify the latent variable's contribution in "explaining away" the observation, we compared the variability in the test set and in the conditionally generated images by calculating the mean squared distance between the samples and their respective empirical mean. Here the mean squared distance serves as a proxy for the entropy in each distribution. We found a substantial decrease in the variability from $5.6 \times 10^{-1}$ to $4 \times 10^{-3}$ when the generation is conditioned on the latent variable, indicating that the representation captures most of the variability in the dataset.

To directly probe the semantic structure of the learned representation, we systematically varied two of the generative factors and extracted the corresponding posterior means provided by the inference network. When varying the $x, y$ coordinates of the disk, we found an approximately Cartesian representation in the latent space (Fig. 2D), while varying the background intensity and the $x$ coordinate resulted in an approximately polar representation (Fig. 2E). The singular point in the center corresponds to an image where the background intensity exactly matches the foreground intensity, so the $x, y$ coordinates are unspecified. The shrinking of the spatial encoding with contrast is a reflection of increasing uncertainty in position pulling the posterior mean towards the prior. We see this more clearly in the 3d space (Fig. 2F), in which we linearly interpolated along all three generative factors. Importantly, the orthogonality of the semantic axes directly reflects the independence of the three generative factors. The constraints imposed by the KL regularization term suffice for the model to learned a factorized, semantically meaningful representation of the data.

**CelebA dataset.** Next, we applied our method to CelebA, a dataset of diverse, high-quality images of human faces with structured attributes (e.g. facial features, expressions, and accessories) that are qualitatively discernible by human observers, which provides sematic structure that we can post-hoc evaluate in the learned representations. Following Ho et al. (2020), we used a standard UNet architecture for the denoiser, but without self-attention layers, so as to focus on the algorithmic effects. The noise level was encoded using a sinusoidal position embedding, provided as an additional input

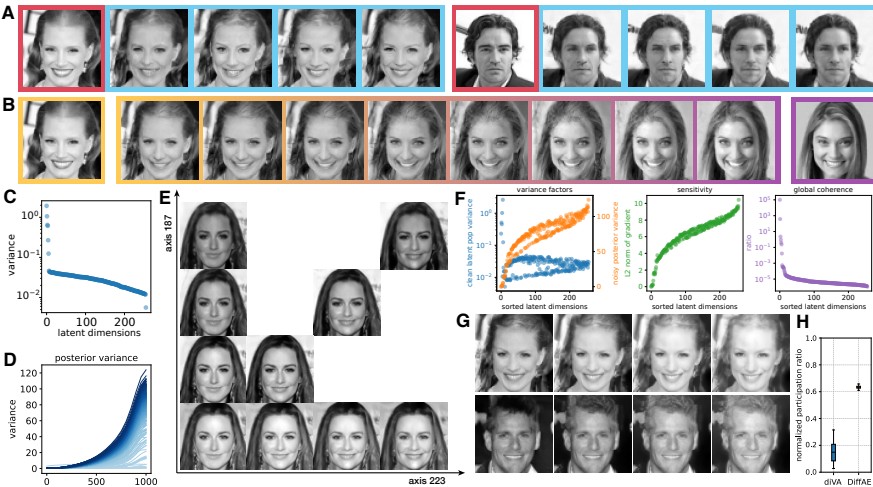

Figure 3: **CelebA. A)** Conditional image generation; conditioning image – red, samples – blue. **B)** Linear traversal between latents of leftmost (yellow) and rightmost (purple) images. **C)** The spectrum of the posterior covariance for one example image from the test set. **D)** The posterior variance as a function of noise level; color intensity marks the ordering of the axes at the largest noise level. **E)** Moving along the linear combination of two latent axes with identified semantics. **F)** Geometric characterization of latent representation. Left: across-test data variability in posterior mean (blue) and posterior variance of a noisy image (orange) for each latent dimension. Middle: norm of the sensitivity of posterior variance to input changes. Right: Global coherence of each axis. The x axes of all three plots are sorted by global coherence. **G)** Example transformation of two different images along a latent axis identified as global. **H)** Weight sparsity of binary classifiers compares how closely the semantics of latent axes align to supervised labels for SAMI vs. DiffAE.

channel (Nichol and Dhariwal, 2021). For the inference network, we used a half-UNet architecture that retains the Down and Mid, but not the Up blocks. The mean and covariance of the posterior are linearly decoded from this network. Unlike Mittal et al. (2023), our inference network does not receive noise level information, which we found to be unnecessary for learning a good representation. As is common in VAEs, we restricted the posterior covariance to be diagonal, $\Sigma_\phi = \mathrm{diag}(\boldsymbol{\sigma}_\phi^2)$. Additional architectural details are provided in Appendix A.5.

We trained the joint model from scratch using 60,000 images from the CelebA dataset, and evaluated the learned representation by assessing conditioned reconstruction quality on images from the test set. If the learned latents are useful, they should encode the pertinent features in the original guiding image that, when used to condition the generation process, reduces the MSE between itself and the generated images. Note that proper conditional generation is not trivial: since guidance scores are added to the denoiser estimates in pixel space, conditioning can easily worsen generation if the latents are non-informative or misaligned with the image prior.

The generated images possess many of the same features as the guiding image, including skin tone, light position, pose, and location and shape of facial features such as the eyes, hairline, and mouth (Fig. 3A). As with the disks, we observe variability in conditionally generated image features, stemming from uncertainty in the posterior and the stochastic nature of the diffusion process. Again, we measured the effect of the latent conditioning and found that the variability is reduced from $6.62 \times 10^{-2}$ to $2.04 \times 10^{-3}$ (full histogram in Appendix A.6). To measure sample fidelity and diversity, we computed the FID based on 10,000 images (Heusel et al., 2017). SAMI's score of 16.25 beats other diffusion-based representation learning models such as DiffAE and InfoDiffusion (Preechakul et al., 2022; Wang et al., 2023) (see Appendix A.7 for details).

VAEs exhibit so-called *latent holes*, i.e. discontinuities in the latent space, that emerge from a mismatch between the approximate posterior and latent prior. These holes reduce the capacity of the latent space and negatively affect generative performance and log likelihoods (Li et al., 2021; Xu et al., 2020; Falorsi et al., 2018). While past work account for these holes explicitly through regularization or by enforcing compactness Zhang et al. (2022a); Glazunov and Zarras (2023), we prove that training

the encoder to satisfy Eq. 10 using both clean images and their noisy counterparts implicitly smooths the latent space by minimizing the expected Frobenius norm of the encoder Hessian taken over all noise levels (see Appendix B.1 for full proof). To investigate the smoothness of the latent embedding we used latent traversal, linearly interpolating between the latent representations corresponding to two images from the test set (Fig. 3B). At regular intervals along this interpolant, we generated sample images conditioned on the corresponding latent. Given two endpoint images (enclosed in yellow and purple boxes in Fig. 3B), the intermediate images display semantic characteristics that change smoothly between those of the endpoint images, indicating that the learned latent space is smooth and continuous.

**Factorized semantics.** The results on the disk images suggest SAMI latents can capture independent generative factors in the data. Indeed, by leveraging the additive nature of the score-based guidance, we can show that a disentangled representation of hierarchical semantic features naturally emerges from this setup. Specifically, under mild assumptions that semantic features are distributed hierarchically with unique characteristic spatial scales, we prove that a diffusion-based generative model optimized to minimize both reconstruction error and the Kullback-Leibler (KL) divergence between a diagonal covariance posterior and an isotropic Gaussian prior yields this disentangled representation (see Appendix B.2 for full proof).

Empirically, the posterior variance for any given image exhibits a few dominant modes alongside many smaller ones (Fig. 3C), indicating that, within the local neighborhood of the mean latent representation, certain latent axes have significantly higher uncertainty. Our explicitly probabilistic approach point to the structure of the posterior covariance as a natural lens into the interpretability of individual latent axes, and accordingly, adding noise to the image increases variance across these axes (Fig. 3D), suggesting that posterior variance reflects the inference network's uncertainty about the corresponding features.

We hypothesize that axes with low posterior variance at low noise levels encode robust semantic features, while the rate of variance increase with noise level indicates the spatial scale of the encoded features. Specifically, large-scale features, still prominent at relatively high noise levels, should correspond to axes with low uncertainty even under significant noise, whereas small-scale features will exhibit overall higher uncertainty.

To test this, we perturbed a clean test image's latent representation along axes with low posterior variance. The resulting images showed consistent changes in specific semantic attributes, suggesting that the model factorizes representations semantically (Fig. 3E, smiling horizontally, forward lighting vertically). Interpolating along a linear combination of two axes produced images reflecting both attributes, suggesting a combinatorial code (Fig. 3E, diagonal). Meanwhile, axes that displayed large changes in posterior variance tended to correspond to finer scale features, such as movements of the eyes and mouth. Finally, we measured disentanglement using Total AUROC Difference (TAD) (Yeats et al., 2022). When trained on CelebA, SAMI attains a TAD score of $0.583$ and captures 3 attributes, which significantly outperforms most diffusion-based methods and VAE-based baselines, and has comparable scores to the state-of-the-art EncDiff model (TAD of $0.638$ but no reported attributes captured) (Yang et al., 2024). We provide full comparisons in Appendix A.7. These findings reveal that SAMI's latent space is structured to disentangle semantic features along specific axes, offering a foundation for interpretable and controllable generative modeling.

**How global is the mapping between latent axes and semantic meaning?** While perturbation along individual axes is interpretable in the neighborhood of the conditioning image, we found that in general this mapping is not preserved across the full latent space. To identify the axes whose semantic meaning is preserved globally, we derived a formula that provides sufficient conditions for global axes under additive transformations by assessing the $\mathbf{z}$-independence of the linear local approximation of the inference network (see Appendix A.8). When applying this metric (Fig. 3F, right) we find the axes with the largest measure to be "junk" dimensions that encode the pixel-level noise, whose semantic mapping is in effect global. However, most of the identified global axes are interpretable, reflecting e.g. global lightning effects (Fig. 3G).

**Alignment with explicit labels.** A separate way to assess the semantic structure of the representation is by measuring the alignment between latent axes and recognizable semantic labels. To this end, we trained separate logistic regression classifiers on the learned representation to predict each of the 40 binary semantic labels. If the axes are well aligned with the supervised attributes, the weights of

the classifier should be sparse, as only a small subset of the axes need to be recruited to form the decision boundary. We used the normalized participation ratio (PR) of each classifier's coefficients as a measure of sparsity; this is 1 if only one latent axis is used for a decision, and it becomes the square root of the dimensionality of the latent space if all axes are used equally. The normalized PR is generally low for SAMI, in particular when compared to state of the art alternative DiffAE (Fig 3H), suggesting that our approach is superior in factorizing the latent space into useful semantic axes.

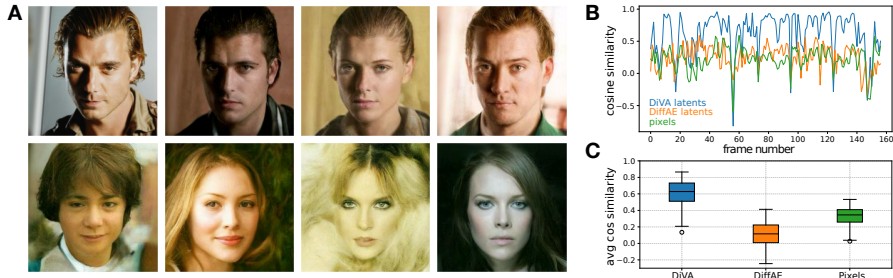

Figure 4: **Unsupervised extraction of latent representation from a pre-trained diffusion model. A)** Top: Samples from the trained model, conditioned on leftmost image (green). Bottom: samples drawn from the unconditional diffusion model. **B)** Straightness of latent trajectories over the course of one naturalistic video from CelebV. **C)** Average straightness of latent encoding (over 50 naturalistic videos) for SAMI, compared to trajectories in pixel space and latent trajectories of DiffAE.

**Feature extraction for pretrained diffusion models.** Since the denoising and guidance terms are parameterized by separate neural networks, one might sensibly wonder if training them jointly is strictly necessary or if the inference network could be trained separately on top of a pre-trained denoiser. To test this, we used a DDPM from the Huggingface repository pretrained on CelebA-HQ-256 (Ho et al., 2020; Karras et al., 2018) and then trained the inference network to guide the responses of the denoiser towards the corresponding clean image using the same objective as before. We found that samples drawn from the conditioned distribution exhibit similar semantic characteristics as the guiding image, such as lighting direction, facial expression, and hairline, while remaining a naturalistic image with high frequency details (Fig. 4). As before, the conditioning reduces variability compared to samples from the unconditional model (Fig. 4A, second row), but there is still appreciable diversity among the samples. Overall, separate training of the conditioning network seems to generally preserve the properties seen with joint training with a substantial reduction in training effort.

**Encoding of video trajectories.** One property that makes for good latent representations is their predictability over time, measured by the ability to represent temporal sequences in a manner that supports linear extrapolation (Niu et al., 2024). Since movement along individual latent axes corresponds to naturalistic transformations in images (Fig. 3E), we hypothesized that SAMI may map natural movements in naturalistic videos, such as a mouth opening and closing, to relatively straight trajectories in latent space. To test this, we encoded frames from a naturalistic facial attributes video dataset (Yu et al., 2023) into latent representations using the SAMI model for CelebA-HQ. Following Hénaff et al. (2019), we quantified the "straightness" of temporal sequences in both latent and pixel domains by computing discrete curvature, defined as the cosine similarity between vectors connecting consecutive pairs of frames in the sequence: $s = \mathbf{d}_t \cdot \mathbf{d}_{t+1} / (\|\mathbf{d}_t\| \|\mathbf{d}_{t+1}\|)$, where $\mathbf{d}_t = \mathbf{f}_{t+1} - \mathbf{f}_t$ represents the difference between consecutive frames, or latent posterior means. Despite being trained solely on static images, our model encodes frames of these videos such that their trajectories in the 512-dimensional latent space exhibit near-unity cosine similarity across large portions of a given video (Fig. 4B). Compared to pixel-domain trajectories and those from other diffusion-based representation learning models (Preechakul et al., 2022), SAMI produces significantly straighter latent trajectories. This effect holds consistently across 50 randomly sampled videos from the dataset (Fig. 4C). Thus, by leveraging its semantically factorized representation, SAMI naturally straightens dynamic patterns without temporal supervision.

## 4 DISCUSSION

While recent work has demonstrated the promise of unsupervised representation learning with diffusion models (Fuest et al., 2024), past approaches typically rely on (often deterministic) encoders that approximate the log-likelihood (Preechakul et al., 2022; Mittal et al., 2023; Yang et al., 2023a; Kim et al., 2024), leverage additional information beyond the base dataset (Hudson et al., 2024), or use objectives that deviate from the ELBO by adding specialized regularization terms, e.g. to encourage mutual information (Wang et al., 2023), sparsity (Mittal et al., 2023), or to bound entropy (Yang et al., 2023b). See Appendix A.9 for detailed comparison to prior methods. SAMI diverges from this paradigm by optimizing an exact ELBO that encodes both clean and noisy images into probabilistic latents.

This dual encoding strategy introduces a novel form of implicit regularization: by requiring the inference network to predict the log-likelihood of clean image features while conditioned on noisy observations, SAMI learns to effectively separate signal from noise and calibrate its posterior variance to capture the irreducible uncertainty inherent in the data. The probabilistic nature of these paired observations constrains the latent space geometry, promoting representations where semantically similar images cluster together. Our empirical analysis reveals that this approach provably and empirically mitigates the "latent holes" problem that commonly afflicts variational autoencoders (Rezende and Viola, 2018; Falorsi et al., 2018; Li et al., 2021; Xu et al., 2020; Zhang et al., 2022a; Glazunov and Zarras, 2023), producing perceptually smoother and more coherent latent spaces.

Pairing a powerful generative model with strong factorization assumptions on the posterior and prior encourages disentanglement without sacrificing generative performance. The KL regularization term enforces an information bottleneck by encouraging posterior whitening, which promotes maximum entropy representations in the latent space (Burgess et al., 2018). Crucially, the conditional denoising objective encourages the latents to capture meaningful structure in the observed data by leveraging the implicit multi-scale image prior learned by the diffusion model. We prove by leveraging the geometric properties of our model that disentangled representations of hierarchical semantic features naturally arise from the use of this image prior, potentially explaining why diffusion-based representation models offer superior disentanglement scores compared to traditional VAEs.

Moreover, our approach sidesteps the traditional rate-distortion trade-off in VAEs: rather than compromising reconstruction fidelity to achieve better disentanglement, SAMI maintains perceptually high-quality generations through the expressive power of the diffusion prior, even when the latent representation is heavily regularized. This information theoretic perspective suggests interesting connections to the I-MMSE relation (Guo et al., 2005; Kong et al., 2023), which has recently been used to link conditional diffusion scores with estimates of mutual information (Franzese et al., 2024). Such connections might provide alternative theoretical foundations for understanding representation learning in conditional diffusion models and could potentially allow for more direct optimization of information-theoretic objectives such as InfoMax (Linsker, 1988). We leave the exploration of these connections as an exciting direction for future work. Overall, our results represent an important and mathematically rigorous step towards building learning systems that extract complex semantical structure from data in an unsupervised manner while maintaining competitive generative performance.

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

# A   APPENDIX

The anonymized code repository can be found at: https://anonymous.4open.science/r/diva-D7DA

## A.1   RELATION OF MMSE SOLUTION TO CONDITIONAL SCORE

Here, we provide a derivation of how the guidance score relates to the MMSE solution for the noise present in the noisy image (Eq. 6). For a variance-preserving noising operator $p(\mathbf{x}_t|\mathbf{x}_0) = \mathcal{N}(\sqrt{\bar{\alpha}_t}\mathbf{x}_0, (1 - \bar{\alpha}_t)I)$, Miyasawa/Tweedie's formula is given as

$$\mathbb{E}_{p(\mathbf{x}_0|\mathbf{x}_t,\mathbf{z})}[\mathbf{x}_0] = \frac{1}{\sqrt{\bar{\alpha}_t}}[\mathbf{x}_t + (1 - \bar{\alpha}_t)\nabla_{\mathbf{x}_t}\log p(\mathbf{x}_t|\mathbf{z})].$$

We can compute the expected value of the noise in the noisy image

$$\mathbb{E}_{p(\mathbf{x}_0|\mathbf{x}_t,\mathbf{z})}[\boldsymbol{\epsilon}] = \int \boldsymbol{\epsilon}\, p(\mathbf{x}_0|\mathbf{x}_t, \mathbf{z})d\mathbf{x}_0$$

$$= \frac{1}{\sqrt{1 - \bar{\alpha}_t}}\left[\mathbf{x}_t - \sqrt{\bar{\alpha}_t}\mathbb{E}_{p(\mathbf{x}_0|\mathbf{x}_t,\mathbf{z})}[\mathbf{x}_0]\right].$$

where on the second line we have substituted in $\boldsymbol{\epsilon} = {}^1/\sqrt{1-\bar{\alpha}_t}(\mathbf{x}_t - \sqrt{\bar{\alpha}_t}\mathbf{x}_0)$ via the re-parameterization trick. Now, rearranging for and substituting the expected value of $\mathbf{x}_0$ into Tweedie's formula above, we are left with

$$\mathbb{E}_{p(\mathbf{x}_0|\mathbf{x}_t,\mathbf{z})}[\boldsymbol{\epsilon}] = \frac{1}{\sqrt{1 - \bar{\alpha}_t}}\left[\mathbf{x}_t - \sqrt{\bar{\alpha}_t}\left(\frac{1}{\sqrt{\bar{\alpha}_t}}[\mathbf{x}_t + (1 - \bar{\alpha}_t)\nabla_{\mathbf{x}_t}\log p(\mathbf{x}_t|\mathbf{z})]\right)\right]$$

$$= \frac{1}{\sqrt{1 - \bar{\alpha}_t}}\left[\mathbf{x}_t - (\mathbf{x}_t + (1 - \bar{\alpha}_t)\nabla_{\mathbf{x}_t}\log p(\mathbf{x}_t|\mathbf{z}))\right]$$

$$= \frac{1}{\sqrt{1 - \bar{\alpha}_t}}\left(-(1 - \bar{\alpha}_t)\nabla_{\mathbf{x}_t}\log p(\mathbf{x}_t|\mathbf{z})\right)$$

$$= -\sqrt{1 - \bar{\alpha}_t}\,\nabla_{\mathbf{x}_t}\log p(\mathbf{x}_t|\mathbf{z})$$

which is the relation given by Eq. 6 in the main text.

## A.2   GUIDING TRANSITION OPERATORS

Here we derive the equation for how the guidance score affects the DDPM reverse transition operator, given by $p(\mathbf{x}_t|\mathbf{x}_{t+1})$. The classifier guidance equation in Eq. 7 can be expressed in terms of the transition operators:

$$\nabla_{\mathbf{x}_t}\log p(\mathbf{x}_t|\mathbf{x}_{t+1}, \mathbf{z}) = \nabla_{\mathbf{x}_t}\log p(\mathbf{x}_t|\mathbf{x}_{t+1}) + \nabla_{\mathbf{x}_t}\log p(\mathbf{z}|\mathbf{x}_t, \mathbf{x}_{t+1})$$
$$= \nabla_{\mathbf{x}_t}\log p(\mathbf{x}_t|\mathbf{x}_{t+1}) + \nabla_{\mathbf{x}_t}\log p(\mathbf{z}|\mathbf{x}_t),$$

where we have used the fact that since $\mathbf{x}_{t+1}$ is a noisier image than $\mathbf{x}_t$, it carries less information about $\mathbf{z}$ than $\mathbf{x}_t$, so the dependence on $\mathbf{x}_{t+1}$ can be dropped. In practice, this problem is equivalent to asking how to sample from $p(\mathbf{x}_{t-1}|\mathbf{x}_t, \mathbf{z})$ over all noise levels $t$. For a given noise schedule determined by $\{\alpha_t\}$, we evaluate $\mathbf{x}_t \sim p_\theta(\mathbf{x}_{t-1}|\mathbf{x}_t) = \mathcal{N}(\boldsymbol{\mu}_\theta, \sqrt{1 - \alpha_t}I)$ by estimating the transition operator mean

$$\boldsymbol{\mu}_\theta(\mathbf{x}_t) = \frac{1}{\sqrt{\alpha_t}}\left(\mathbf{x}_t - \frac{1 - \alpha_t}{\sqrt{1 - \bar{\alpha}_t}}\boldsymbol{\epsilon}_\theta(\mathbf{x}_t)\right),$$

using which we can estimate the next state in the Markov chain $\mathbf{x}_{t-1} = \boldsymbol{\mu}_\theta(\mathbf{x}_t) + \sqrt{1 - \alpha_t}\,\boldsymbol{\epsilon}$, where $\boldsymbol{\epsilon} \sim \mathcal{N}(0, I)$. To sample from the $\mathbf{z}$-conditional distribution instead, this function must be dependent

on $\boldsymbol{\epsilon}_{\theta,\phi}(\mathbf{x}_t, \mathbf{z})$ rather than on $\boldsymbol{\epsilon}_\theta(\mathbf{x}_t)$:

$$\boldsymbol{\mu}(\mathbf{x}_t, \mathbf{z}) = \frac{1}{\sqrt{\alpha_t}}\left(\mathbf{x}_t - \frac{1-\alpha_t}{\sqrt{1-\bar{\alpha}_t}}\boldsymbol{\epsilon}_{\theta,\phi}(\mathbf{x}_t, \mathbf{z})\right)$$

$$= \frac{1}{\sqrt{\alpha_t}}\left(\mathbf{x}_t - \frac{1-\alpha_t}{\sqrt{1-\bar{\alpha}_t}}\left(\boldsymbol{\epsilon}_\theta(\mathbf{x}_t) - \sqrt{1-\bar{\alpha}_t}\nabla_{\mathbf{x}_t}\log q_\phi(\mathbf{z}|\mathbf{x}_t)\right)\right)$$

$$= \frac{1}{\sqrt{\alpha_t}}\left(\mathbf{x}_t - \frac{1-\alpha_t}{\sqrt{1-\bar{\alpha}_t}}\boldsymbol{\epsilon}_\theta(\mathbf{x}_t)\right) - \frac{1-\alpha_t}{\sqrt{\alpha_t}}\nabla_{\mathbf{x}_t}\log q_\phi(\mathbf{z}|\mathbf{x}_t)$$

$$= \boldsymbol{\mu}_\theta(\mathbf{x}_t) - \frac{1-\alpha_t}{\sqrt{\alpha_t}}\nabla_{\mathbf{x}_t}\log q_\phi(\mathbf{z}|\mathbf{x}_t).$$

Since $\mathbf{x}_{t-1}(\mathbf{x}_t, \mathbf{z}) = \boldsymbol{\mu}(\mathbf{x}_t, \mathbf{z}) + \sqrt{1-\alpha_t}\boldsymbol{\epsilon}$, we can rewrite this relation in terms of the noisy states:

$$\mathbf{x}_{t-1}^*(\mathbf{x}_t, \mathbf{z}) = \mathbf{x}_{t-1}^*(\mathbf{x}_t) - \frac{1-\alpha_t}{\sqrt{\alpha_t}}\nabla_{\mathbf{x}_t}\log q_\phi(\mathbf{z}|\mathbf{x}_t)$$

$$= \mathbf{x}_{t-1}^*(\mathbf{x}_t) - \frac{1-\alpha_t}{\sqrt{\alpha_t}}\mathbf{g}_{t,\phi}.$$

## A.3 TRAINING AND CONDITIONAL GENERATION ALGORITHMS

Below, we provide algorithms for training SAMI and for performing conditional generation using the trained networks.

---

**Algorithm 1:** Conditional generation, given a clean guidance image $\mathbf{x}_0$, trained denoiser $\boldsymbol{\epsilon}_\theta$ and inference network $f_\phi$

---

$(\boldsymbol{\mu}_0, \boldsymbol{\sigma}_0) \leftarrow f_\phi(\mathbf{x}_0)$                             `// encode guidance image`
$\boldsymbol{\epsilon} \sim \mathcal{N}(0, I)$
$\mathbf{z} \leftarrow \boldsymbol{\mu}_0 + \boldsymbol{\sigma}_0 \cdot \boldsymbol{\epsilon}$                                 `// sample clean image latent`

Initialize $t \leftarrow T$; $\mathbf{x}_t \sim \mathcal{N}(0, I)$
**while** $t \neq 0$ **do**
    $(\boldsymbol{\mu}_t, \boldsymbol{\sigma}_t) \leftarrow f_\phi(\mathbf{x}_t)$                         `// encode noisy image`
    $\log q \leftarrow -1/2\big(d_M(\mathbf{z}, \boldsymbol{\mu}_t) + \log|\boldsymbol{\sigma}_t I|\big)$            `// log posterior`
    $\mathbf{g}_t \leftarrow \nabla_{\mathbf{x}_t}\log q$                            `// guidance score`
    $\boldsymbol{\mu} \leftarrow \frac{1}{\sqrt{\alpha_t}}\left(\mathbf{x}_t - \frac{1-\alpha_t}{\sqrt{1-\bar{\alpha}_t}}\boldsymbol{\epsilon}_\theta\right)$
    $\Sigma \leftarrow \sqrt{1-\alpha_t}I$
    $\mathbf{x}_t \sim \mathcal{N}(\boldsymbol{\mu} + \Sigma\mathbf{g}_t, \Sigma)$                      `// guidance`
    $t \leftarrow t - 1$
**end**

---

## A.4 CONDITIONAL GENERATION AS CONSTRAINED OPTIMIZATION

As mentioned in the "Geometric Intuition" subsection, we can think of conditional generation as a constrained optimization problem, in which we have multiple constraints imposed by the inference network and the denoiser. To make this perspective more concrete, consider the generation procedure, given by Algorithm 1. We start with a sample of white noise, $\mathbf{x}_t \sim \mathcal{N}(0, I)$, from which we take gradient steps towards a region of large guided score:

$$\mathbf{x}_t \leftarrow \mathbf{x}_t + \nabla_{\mathbf{x}_t}\log p(\mathbf{x}_t|\mathbf{z})$$
$$= \mathbf{x}_t + \nabla_{\mathbf{x}_t}\log q_\phi(\mathbf{z}|\mathbf{x}_t) + \nabla_{\mathbf{x}_t}\log p_\theta(\mathbf{x}_t),$$

where we have decomposed the conditional score using Bayes' rule. Since the posterior $q_\phi$ is assumed to be conditional Gaussian, the analytic form of the log likelihood is given as

$$\log \mathscr{L}_\phi(\mathbf{x}_t; \mathbf{z}) = -\frac{1}{2}\,\mathbb{E}_{\mathbf{z}'\sim q_\phi(\mathbf{z}|\mathbf{x}_0)}\Big[d_M\big(\mathbf{z}'_\phi(\mathbf{x}_0), \boldsymbol{\mu}_\phi(\mathbf{x}_t)\big) + \log|\Sigma_\phi(\mathbf{x}_t)| + c\Big], \qquad (11)$$

**Algorithm 2:** Training, given a denoiser network $\epsilon_\theta$ and inference network $f_\phi$ with randomly initialized weights

---

**while** *not converged* **do**

$\quad \mathbf{x}_0 \sim p(\mathbf{x}_0)$

$\quad t \sim [0, T]$

$\quad \mathbf{x}_t \sim p_{\text{fwd}}(\mathbf{x}_t|\mathbf{x}_0) = \mathcal{N}(\sqrt{\bar{\alpha}_t}\mathbf{x}_0, (1 - \bar{\alpha}_t)I)$        // noise the image

$\quad \boldsymbol{\epsilon}_\mathbf{x} \leftarrow (\mathbf{x}_t - \sqrt{\bar{\alpha}_t}\mathbf{x}_0)/\sqrt{1-\bar{\alpha}_t}$

$\quad \boldsymbol{\epsilon}_\mathbf{z} \sim \mathcal{N}(0, I)$

$\quad (\boldsymbol{\mu}_0, \boldsymbol{\sigma}_0) \leftarrow f_\phi(\mathbf{x}_0)$        // encode clean guidance image

$\quad \mathbf{z} \leftarrow \boldsymbol{\mu}_0 + \boldsymbol{\sigma}_0 \cdot \boldsymbol{\epsilon}_\mathbf{z}$

$\quad (\boldsymbol{\mu}_t, \boldsymbol{\sigma}_t) \leftarrow f_\phi(\mathbf{x}_t)$        // encode noisy image

$\quad \log q \leftarrow -1/2(d_M(\mathbf{z}, \boldsymbol{\mu}_t) + \log|\boldsymbol{\sigma}_t I|)$        // log posterior

$\quad \mathbf{g}_{t,\phi}(\mathbf{z}) \leftarrow \nabla_{\mathbf{x}_t} \log q$        // guidance score

$\quad \mathcal{L}_\mathbf{x} \leftarrow \|\boldsymbol{\epsilon}_\mathbf{x} - \boldsymbol{\epsilon}_\theta(\mathbf{x}_t) - \sqrt{1 - \bar{\alpha}_t}\,\mathbf{g}_{t,\phi}(\mathbf{x}_t, \mathbf{z})\|^2$

$\quad \mathcal{L}_\mathbf{z} \leftarrow D_{KL}(\mathcal{N}(\boldsymbol{\mu}_0, \boldsymbol{\sigma}_0)\|\mathcal{N}(0, I))$

$\quad$ take gradient steps on $\nabla_{\theta,\phi}(\mathcal{L}_\mathbf{x} + \beta\mathcal{L}_\mathbf{z})$

**end**

---

where $d_M = (\mathbf{z}'_\phi(\mathbf{x}_0) - \boldsymbol{\mu}_\phi(\mathbf{x}_t))^\top \Sigma_\phi^{-1}(\mathbf{z}'_\phi(\mathbf{x}_0) - \boldsymbol{\mu}_\phi(\mathbf{x}_t))$ is the Mahalanobis distance between the mean estimate of the network state in latent space $\boldsymbol{\mu}_\phi(\mathbf{x}_t)$ and the latent corresponding to the target image $\mathbf{z}'(\mathbf{x}_0)$, computed under a metric that is given by the inverse of the covariance $\Sigma_\phi^{-1}$ (red arrow and concentric ellipsoids in Fig. 1C, $\mathcal{X}$ space).

Using this expression for the log likelihood, it becomes clear that we are taking gradient steps with respect to multiple constraints:

$$\mathbf{x}_t \leftarrow \mathbf{x}_t + \nabla_{\mathbf{x}_t} \log q_\phi(\mathbf{z}|\mathbf{x}_t) + \nabla_{\mathbf{x}_t} \log p_\theta(\mathbf{x}_t)$$

$$= \mathbf{x}_t - \frac{1}{2}\nabla_{\mathbf{x}_t}\left(\mathbb{E}_{\mathbf{z}'\sim q_\phi(\mathbf{z}|\mathbf{x}_0)}\Big[d_M(\mathbf{z}', \boldsymbol{\mu}_\phi(\mathbf{x}_t)) + \log|\Sigma_\phi(\mathbf{x}_t)|\Big]\right) + \nabla_{\mathbf{x}_t} \log p_\theta(\mathbf{x}_t)$$

$$= \mathbf{x}_t - \frac{1}{2}\mathbb{E}_{\mathbf{z}'\sim q_\phi(\mathbf{z}|\mathbf{x}_0)}\Big[\nabla_{\mathbf{x}_t}d_M(\mathbf{z}', \boldsymbol{\mu}_\phi(\mathbf{x}_t))\Big] - \frac{1}{2}\nabla_{\mathbf{x}_t}\log|\Sigma_\phi(\mathbf{x}_t)| + \nabla_{\mathbf{x}_t}\log p_\theta(\mathbf{x}_t).$$

There are three terms, each corresponding to a specific constraint during this optimization procedure. The first constraint is to land in a region of the image space such that its representation minimizes its Mahalanobis distance to the latent of the guiding image $\mathbf{z}'(\mathbf{x}_0)$. This is related to the metamer perspective (Helmholtz, 1852; Zhu et al., 1998; Portilla and Simoncelli, 2000; Freeman and Simoncelli, 2011; Mahendran and Vedaldi, 2014; Feather et al., 2023), where the goal is to produce data samples that share the same latent representation as a target datum. Metamer generation can be formulated as a constrained optimization problem, minimizing the L2 distance between the representations of the current sample and the target, $\mathbf{x}_t \leftarrow \mathbf{x}_t - \nabla_{\mathbf{x}_t}\|\mathbf{r}_{\text{target}} - \mathbf{r}_{\mathbf{x}_t}\|^2$, where $\mathbf{r}_{\mathbf{x}_t}$ is the representation of the current sample $\mathbf{x}_t$, and $\mathbf{r}_{\text{target}}$ that the target datum. Our approach generalizes this framework by replacing the L2 distance with the log likelihood from Eq. 11. In this view, the inference network constrains the generation process to produce samples that are metameric to the guidance image as defined by the latent representation.

The next term corresponds to taking steps towards regions of the image space that minimizes the log determinant of the covariance matrix. Geometrically, we can think of this as reducing the volume of the uncertainty ellipsoid over the features identified by the inference network. In a well-calibrated Bayesian network, the posterior uncertainty over the identified features should match the irreducible uncertainty present in the image. For SAMI, the irreducible uncertainty in the latent space comes from the noise in the noisy observations, and as such we should expect that the log determinant of the covariance to correlate strongly with the variance of the noisy image.

The final term corresponds to the denoiser, parameterized by $\theta$, which ensures that the network is driven towards the natural image manifold, i.e. the set of highly probable images.

The region of the $\mathcal{X}$ space that we land on is at the intersection of these three signals. This perspective suggests that we can increase the diversity (i.e. the entropy) of the generated images by relaxing

one or more of these constraints. For example, if the generation procedure were given only by the third term, the network would be driven to satisfy only the image prior, recovering the unconditional generation of DDPM. Alternatively, we might choose to calculate the Mahalanobis distance over only a subset of the latent dimensions, in effect making sure that $\boldsymbol{\mu}_\phi$ is close to $\mathbf{z}'$ only in those latent dimensions. This has the effect of making those latent axes "rigid", while the rest of the axes are "sloppy", since large distances in these dimensions are discounted Machta et al. (2013); Transtrum et al. (2015). Back in the pixel space, this equates to finding images that possess a subset of the semantic attributes in the original guiding image, i.e. those that correspond to the constrained feature dimensions.

## A.5 ARCHITECTURE AND TRAINING DETAILS

| Parameter | Disks | CelebA 64 | CelebA-HQ 256 |
|---|---|---|---|
| Inference net arch. | ConvNet | Half-UNet | Half-UNet |
| Infnet base channels | 48 | 64 | 64 |
| Infnet channel multiplier | [2, 2] | [1, 1, 1, 1, 1, 1] | [2, 2, 2, 2, 1, 1] |
| Infnet nonlinearity | ReLU | ReLU | SiLU |
| Denoiser arch. | UNet (no attn) | UNet (no attn) | UNet (with attn, frozen) |
| Denoiser base channels | 128 | 128 | 128 |
| Denoiser ch. multiplier | [1, 1, 1, 1, 1, 1] | [1, 1, 1, 1, 1, 1] | [1, 1, 2, 2, 4, 4] |
| Num. noise levels | 400 | 1000 | 1000 |
| Latent dimensionality | 3 | 256 | 512 |
| KL weight | 5e-6 | 1e-3 | 1e-8 |
| Noise schedule | Linear | Cosine | Linear |
| Training set size | 2000 | 60,000 | 30,000 |
| Batch size | 512 | 512 | 512 |
| Num. epochs trained | 1000 | 5000 | 70 |
| Learning rate | 6e-3 | 1e-3 | 3e-5 |
| Timestep sampling dist. | Uniform | Uniform | Monotonically increasing |
| Resources (num. A100) | 1 | 20 | 20 |
| Optimizer | Adam (without weight decay) | | |

Table 1: Architectural and training details for each of the three datasets mentioned in the main text.

**Architectural details.**   We tried to use simple architectures throughout our experiments for the sake of interpretability and to emphasize the algorithmic effects as much as possible. For the disks dataset, we used a two layer ConvNet as the inference network, with kernel size 3 and stride 2. The inference networks used for the disks and CelebA-64 datasets have no biases, following Mohan et al. (2022), and the denoisers had no attention blocks. Despite the deliberate restrictions in architecture, we saw that the model was able to learn a robust representation of the dataset. The Half-UNet architecture is a UNet where the Up blocks have been replaced by two MLP layers, both outputting a vector of dimension equal to the latent dimension. From one MLP we get the latent mean. The other outputs a vector which we apply Softplus and squaring operators; this returns the variance.

For the CelebA-HQ 256 dataset, we used an off the shelf DDPM Ho et al. (2020). We froze the denoiser's weights and trained only the inference network, which we again restricted to be a Half-UNet without attention blocks.

**KL weight.**   We annealed the KL weight to the value given in Table 1, starting from a number many orders of magnitude smaller and using a exponential annealing schedule.

**Choosing hyperparameters.**   We performed an extensive grid search over the space of hyperparameters for each model, measuring model performance by the change in MSE between the MSE of an unconditional diffusion model and the MSE when the inference network was also employed.

The unconditional diffusion model retained the same denoiser hyperparameters as the full SAMI. As mentioned in the main text, an informative representation should result in generated images that resemble the original guiding image, and thus a smaller MSE can be used as a measure of latent informativeness. We found that larger KL weight corresponded to more factorized representations, but the weight could only be increased up to a threshold, above which the model exhibits posterior collapse and the representation becomes uninformative. As such, we performed binary search over the KL weights until we found a value just smaller than the threshold.

**Timestep sampling distribution.** When training the model on the disks and CelebA 64 datasets, we sampled the noise level uniformly from 0 to T, as is common when training unconditional DDPMs. When training the inference network on top of the frozen denoiser on the CelebA-HQ dataset, we found that oversampling the larger noise levels was more effective. This is equivalent to over-weighing the contribution of high noise level terms in the overall MSE loss rather than weighing all levels uniformly. If the inference network is unable to learn the guidance vector for all noise levels when paired with a frozen denoiser, since the guidance vector has greater relative influence at high noise levels compared to the denoising score, it is more effective to learn the guidance score for the high noise regime.

**Bottleneck size.** We found that the number of semantic attributes learned by the SAMI representation depended on three factors: the magnitude of the KL weight, the dimensionality of the latent representation, and the timestep sampling distribution. Relaxing the first two of these three factors increases the amount of information that can be learned by the network, but results in a less factorized and interpretable representation.

A.6    REDUCTION IN VARIABILITY FROM CONDITIONING

As a measure of how much entropy is reduced by conditioning the diffusion process on the representation, we measured the entropy in the test set and and the entropy in conditionally generated samples. Since it is not possible to get an exact measure of entropy, we make a coarse Gaussian assumption on the distribution and measure the mean squared distance of samples from their empirical mean. We conditionally generate 4100 samples per conditioning image, which is greater than the ambient dimensionality of the image of 4096. We find that the mean distance is reduced from $6.62 \times 10^{-2}$ to $2.04 \times 10^{-3}$, indicating that the representation captures much of the potential variability in the dataset. We also find a reduction in the spread of the squared distances, from a SEM value of $4.85 \times 10^{-4}$ to $4.45 \times 10^{-6}$. This concentration of the squared distances around a near-zero mean value suggests that there is very little variability in conditionally generated samples.

A.7    METRICS

For disentanglement, we measured the degree to which SAMI's representation was disentangled when trained on CelebA $64 \times 64$. We used the Total AUROC Difference (TAD) metric, which measures the degree to which a unique attribute is detected by a unique latent representation, and the degree to which the latent axes confidently replicate the ground truth axes (Yeats et al., 2022). Among the diffusion-based representation learning methods, InfoDiff, DisDiff and DBAE are trained to exhibit disentangled representations (Wang et al., 2023; Yang et al., 2023b; Kim et al., 2024), so direct comparison against these methods are particularly interesting. We also compared our method against more standard benchmarks in the disentanglement literature, such as VAE and $\beta$-VAE.

In Table 2, we see that though we performed little to no hyperparameter search, SAMI performs the best or second best at both disentangling and sample quality metrics. This is likely due to the effect of training our model to produce the exact log likelihood, which requires both clean and noisy images, rather than a representation of only clean images. The coding of noisy images adds important structure to the latent space by encouraging noisy images to have the same latent value as clean images in latent dimensions that code for coarse features. This ensures that coarser features have more global coherence, a hypothesis that is borne out in Fig. 3F and G.

**dSprites** We also trained our model on the dSprites dataset (Higgins et al., 2017), to see if the disentanglement properties we find in the synthetic disks dataset (Fig. 2) generalize to other synthetic benchmarks where the ground truth factors are known. Of the many disentanglement metrics,

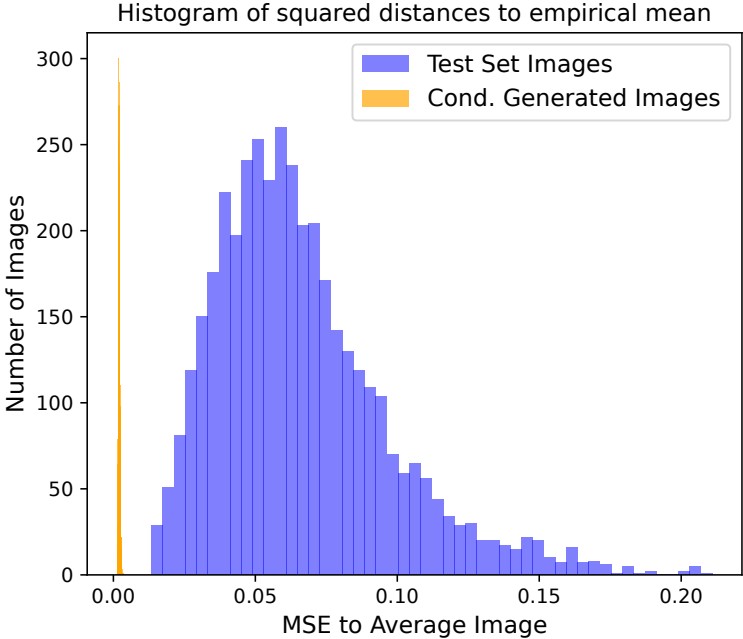

Figure 5: Reduction in variance from conditioning on the posterior on CelebA.

| Model | TAD↑ | Capt. attrs.↑ | FID↓ |
|---|---|---|---|
| AE | $0.042 \pm 0.004$ | $1.0 \pm 0.0$ | $90.4 \pm 1.8$ |
| VAE | $0.000 \pm 0.000$ | $0.0 \pm 0.0$ | $94.3 \pm 2.8$ |
| $\beta$-VAE | $0.088 \pm 0.051$ | $1.6 \pm 0.8$ | $99.8 \pm 2.4$ |
| InfoVAE | $0.000 \pm 0.000$ | $0.0 \pm 0.0$ | $77.8 \pm 1.6$ |
| DiffAE (Preechakul et al., 2022) | $0.155 \pm 0.010$ | $2.0 \pm 0.0$ | $22.7 \pm 2.1$ |
| InfoDiffusion (Wang et al., 2023) | $0.299 \pm 0.006$ | $3.0 \pm 0.0$ | $22.3 \pm 1.2$ |
| DisDiff (Yang et al., 2023b) | $0.305 \pm 0.010$ | - | $18.2 \pm 2.1$ |
| DBAE + TC (Kim et al., 2024) | $0.362 \pm 0.036$ | $3.8 \pm 0.8$ | $13.4 \pm 0.2$ |
| EncDiff (Yang et al., 2024) | $0.638 \pm 0.008$ | - | $14.8 \pm 2.3$ |
| SAMI (ours) | $0.583$ | $3.0$ | $16.3$ |

Table 2: Comparisons against other generative models on TAD disentanglement metric and FID scores on $64 \times 64$ CelebA. "Capt. attrs." refers to the number of captured attributes when calculating the TAD score. Dark gray cells indicate the best, while light gray cells indicate second best.

we measured Modularity (Ridgeway and Mozer, 2018) and DCI Disentanglement (Eastwood and Williams, 2018), since Locatello et al. (2019) finds that all metrics except Modularity are strongly correlated on the dSprites dataset. We measured a Modularity score of $0.882 \pm 0.003$ and a DCI Disentanglement score of $0.117 \pm 0.0005$, after initializing the measurement on 10 separate seeds. For both metrics we used $10\,000$ training samples and $5000$ test samples.

These measures are comparable with the mean disentanglement scores produced by multiple VAE-based models, as shown in Fig. 14, top row in (Locatello et al., 2019). This suggests that our model learns a representation that disentangles ground truth factors despite the fact that our model does not have explicit regularization terms in the objective that encourage disentanglement beyond the $\beta$ parameter, unlike (Chen et al., 2019; Kumar et al., 2018). Moreover, this result indicates that the disentanglement properties of our model still hold despite the fact that the multiscale inductive biases

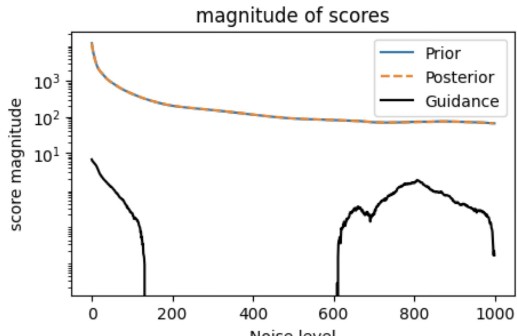

Figure 6: Magnitude of the prior and posterior scores during a single conditional generation procedure, and their difference.

of our model are not aligned with the dSprites dataset, which compresses most of its variability within a smaller range of spatial scales than does natural datasets such as CelebA.

Indeed, we can measure the degree of variability in the dataset at different spatial scales by measuring the difference in magnitude between the prior and posterior scores at each noise level during the conditional generation process. A large magnitude at a particular noise level indicates how much image-space variability the latents are explaining at each noise level, suggesting a specific feature has become visible to the inference network. Conversely, small magnitudes are an indication that at that noise level there are no distinguishing features beyond the posterior mean (which is given by the unconditional denoiser).

We find that there are consistently two "bumps" in magnitude during the conditional denoising process, suggesting that most of the features in the target sprite can be found at two regions of spatial scales. This shows that a disentangled representation is possible, even when the distribution of the features' "characteristic noise levels" (as defined in Appendix B.2) are not spread out evenly across all noise levels, as in natural images.

A.8  DERIVATION FOR GLOBALLY PRESERVED SEMANTIC AXES

We investigated the locality of the mapping between the axes and the semantic label by perturbing the latents of two different images along the same axis. We observed that though in general, this mapping was not preserved across different regions of the latent space, some axes resulted in relatively similar semantic transformations irrespective of the latent.

To identify latent-semantic mappings that are relatively global, we derived a mathematical formulation that provides sufficient conditions for a mapping to be global. Suppose we have a trained conditional diffusion process that maps noisy images $\mathbf{x}_t$ to a clean image estimate $\hat{\mathbf{x}}_0$ via the use of a set of learned latents $\mathbf{z}$. This mapping is defined as a function $f_{\theta,\phi}$ that takes in $\mathbf{z}$ and $\mathbf{x}_t$ and returns a guided clean image estimate:

$$f_{\theta,\phi}(\mathbf{x}_t, \mathbf{z}) = \mathbb{E}_{p_\theta(\mathbf{x}_0|\mathbf{x}_t,\, \mathbf{z})}\big[\mathbf{x}_0\big]$$

$$= \mathbb{E}_{p_\theta(\mathbf{x}_0|\mathbf{x}_t)}\big[\mathbf{x}_0\big] + \frac{1-\bar{\alpha}_t}{\sqrt{\bar{\alpha}_t}}\nabla_{\mathbf{x}_t}\log q_\phi(\mathbf{z}|\mathbf{x}_t)$$

$$= \mathbf{x}_0^*(\mathbf{x}_t;\theta) + \gamma_t\,\nabla_{\mathbf{x}_t}\log q_\phi(\mathbf{z}|\mathbf{x}_t) \tag{12}$$

where the analytic form of $\nabla_{\mathbf{x}_t}\log q_\phi(\mathbf{z}|\mathbf{x}_t)$ is given as the derivative of Eq. 11. Given an initial image $\mathbf{x}_0^* = \mathbf{x}_0^*(\mathbf{x}_t, \mathbf{z}')$ that was generated by latent $\mathbf{z}'$, we define the change in the generated image in response to a perturbation to the latent along axis $i$ as a nonlinear function $\psi(\cdot)$ that we can define, without loss of generality, as

$$\mathbf{x}_0^* + \psi(\mathbf{x}_0^*,\, \mathbf{z},\, i) = f(\mathbf{x}_t,\, \mathbf{z} + \alpha\,\hat{\mathbf{i}})$$

where $\alpha\,\hat{\mathbf{i}}$ is a perturbation of magnitude $\alpha$ along a particular axis $i$. Our aim is to understand whether the functional form of the transformation $\psi$ is dependent on the region of latent space that we are

perturbing. In other words, if this image transformation $\psi$ is global, it should be independent of the latent value $\mathbf{z}$, while if the transformation is more local, then the form of $\psi$ should itself change as our test latent point $\mathbf{z}$ changes.

To make this problem tractable, let us assume we are currently in a state with large noise $\mathbf{x}_t$, where $t \approx T$. At high noise levels, the optimal estimate of the clean image that minimizes the mean squared error is the mean of the training set. This suggests that for a well-trained denoiser, we should expect the same output irrespective of the network's input:

$$\hat{\mathbf{x}}_0(\mathbf{x}_t; \theta) = \mathbb{E}_{p(\mathbf{x}_0|\mathbf{x}_t)}\big[\mathbf{x}_0\big] \approx \mathbb{E}_{p_{\text{dataset}}(\mathbf{x}_0)}\big[\mathbf{x}_0\big] = \mathbf{c}.$$

where $\mathbf{c}$ is a constant image vector. Under this assumption, our function $f$ is no longer dependent on the denoiser but only depends on the inference network:

$$f_\phi(\mathbf{x}_t, \mathbf{z}) = \mathbf{c} + \gamma_t \, \nabla_{\mathbf{x}_t} \log q_\phi(\mathbf{z}|\mathbf{x}_t).$$

This means the transformation $\psi$ is also independent of the denoiser. It is purely a function of the guidance vector:

$$\mathbf{x}_0^* + \psi(\mathbf{x}_0^*, \mathbf{z}, i) = \mathbf{c} + \gamma_t \nabla_{\mathbf{x}_t} \log q_\phi(\mathbf{z} + \alpha \hat{\mathbf{i}} | \mathbf{x}_t)$$
$$= \mathbf{c} - \frac{\gamma_t}{2} \nabla_{\mathbf{x}_t} \Big[ \big(\mathbf{z} + \alpha\,\hat{\mathbf{i}} - \boldsymbol{\mu}_\phi\big)^\top \Sigma_\phi^{-1} \big(\mathbf{z} + \alpha\,\hat{\mathbf{i}} - \boldsymbol{\mu}_\phi\big) + \log\big|\Sigma_\phi\big| - N\log(2\pi) \Big],$$

where $\mathbf{z}$ is a function of $\mathbf{x}_0$, and the means $\boldsymbol{\mu}_\phi$ and covariances $\Sigma_\phi$ are functions of the current noisy image $\mathbf{x}_t$. If we decompose the RHS of the above equation into the contribution of just $\mathbf{z}$ and the contribution of $\alpha\,\hat{\mathbf{i}}$, and denote the last two terms as the scalar $s$, we get:

$$\hat{\mathbf{x}}_0^* + \psi(\hat{\mathbf{x}}_0^*, \mathbf{z}, i)$$
$$= \mathbf{c} - \frac{\gamma_t}{2} \nabla_{\mathbf{x}_t} \Big[ \big(\mathbf{z} - \boldsymbol{\mu}_\phi\big)^\top \Sigma_\phi^{-1} \big(\mathbf{z} - \boldsymbol{\mu}_\phi\big) + 2\big(\mathbf{z} - \boldsymbol{\mu}_\phi\big)^\top \Sigma_\phi^{-1}(\alpha\,\hat{\mathbf{i}}) + (\alpha\,\hat{\mathbf{i}})^\top \Sigma_\phi^{-1}(\alpha\,\hat{\mathbf{i}}) + s \Big]$$
$$= \mathbf{c} - \frac{\gamma_t}{2} \nabla_{\mathbf{x}_t} \Big[ \big(\mathbf{z} - \boldsymbol{\mu}_\phi\big)^\top \Sigma_\phi^{-1} \big(\mathbf{z} - \boldsymbol{\mu}_\phi\big) + 2\alpha\big(\mathbf{z} - \boldsymbol{\mu}_\phi\big)^\top (\Sigma_\phi^{-1})_i + \alpha^2 \, (\Sigma_\phi^{-1})_{ii} + s \Big],$$

where $(\Sigma_\phi^{-1})_i$ is the $i$-th column of the inverse covariance matrix, and $(\Sigma_\phi^{-1})_{ii} = \sigma_i^{-2}$ is the $i$-th element along the diagonal of the inverse covariance matrix. We can now substitute $\hat{\mathbf{x}}_0^* = \mathbf{c} - \frac{\gamma_t}{2}\nabla_{\mathbf{x}_t}\Big[\big(\mathbf{z} - \boldsymbol{\mu}_\phi\big)^\top \Sigma_\phi^{-1} \big(\mathbf{z} - \boldsymbol{\mu}_\phi\big) + s\,\Big]$, which yields

$$\psi(\mathbf{z}, i) = \nabla_{\mathbf{x}_t} \Big[ 2\alpha(\mathbf{z} - \boldsymbol{\mu}_\phi)^\top (\Sigma_\phi^{-1})_i + \alpha^2 \, (\Sigma_\phi^{-1})_{ii} \Big].$$

Let's see how this transformation in $\mathcal{X}$ changes with our location $\mathbf{z}$ in latent space $\mathcal{Z}$. First suppose we use the same latent perturbation $\alpha\,\hat{\mathbf{z}}_i$ irrespective of where we are in latent space. For a given state $\mathbf{x}_t$, we compare two instances of $\psi$ at different values of $\mathbf{z}$:

$$\psi_1(\mathbf{z}^{(1)}, i) = \nabla_{\mathbf{x}_t} \Big[ 2\alpha(\mathbf{z}^{(1)} - \boldsymbol{\mu}_\phi)^\top (\Sigma_\phi^{-1})_i + \alpha^2 \, (\Sigma_\phi^{-1})_{ii} \Big]$$
$$\psi_2(\mathbf{z}^{(2)}, i) = \nabla_{\mathbf{x}_t} \Big[ 2\alpha(\mathbf{z}^{(2)} - \boldsymbol{\mu}_\phi)^\top (\Sigma_\phi^{-1})_i + \alpha^2 \, (\Sigma_\phi^{-1})_{ii} \Big].$$

Since $\boldsymbol{\mu}_\phi$ and $\Sigma_\phi$ are functions of a common $\mathbf{x}_t$, their values are the same in both $\psi_1$ and $\psi_2$. The difference between these two transformations is therefore

$$\psi_1(\mathbf{z}^{(1)}) - \psi_2(\mathbf{z}^{(2)}) = \nabla_{\mathbf{x}_t} \Big[ 2\alpha(\mathbf{z}^{(1)} - \boldsymbol{\mu}_\phi)^\top (\Sigma_\phi^{-1})_i - 2\alpha(\mathbf{z}^{(2)} - \boldsymbol{\mu}_\phi)^\top (\Sigma_\phi^{-1})_i \Big]$$
$$= 2\alpha \nabla_{\mathbf{x}_t} \Big[ (\mathbf{z}^{(1)} - \mathbf{z}^{(2)})^\top (\Sigma_\phi^{-1})_i \Big].$$

We see that the difference in the transformation scales linearly with the difference in $\mathbf{z}^{(1)}$ and $\mathbf{z}^{(2)}$, and scales with the $i$-th column of the inverse covariance matrix. If we assume a diagonal covariance assumption, as in our model, this is simply a unit vector scaled by the $i$-th element along the diagonal:

$$\psi_1(\mathbf{z}^{(1)}) - \psi_2(\mathbf{z}^{(2)}) = 2\alpha \, \nabla_{\mathbf{x}_t} \Big[ (\mathbf{z}^{(1)} - \mathbf{z}^{(2)})^\top \sigma_i^{-2} \hat{\mathbf{z}}_i \Big]$$
$$= 2\alpha \big( \nabla_{\mathbf{x}_t} \sigma_i^{-2} \big)\big( \mathbf{z}_i^{(1)} - \mathbf{z}_i^{(2)} \big), \tag{13}$$

where in the second line we use the fact that only $\sigma_i^{-2}$ is a function of $\mathbf{x}_t$. This means that if the clean images are mapped to similar values in this dimension (e.g. if the population variance in this dimension is very low), then the function that is applied in the data space will be very similar. The effect is also scaled by the derivative of the precision: we want to look for axes where the precision isn't affected by perturbations around $\mathbf{x}_t$:

$$\nabla_{\mathbf{x}_t}\sigma_i^{-2}(\mathbf{x}_t) = \frac{\partial}{\partial\sigma_i^2}\left(\frac{1}{\sigma_i^2}\right)\frac{\partial\sigma_i^2}{\partial\mathbf{x}_t} = -\frac{1}{\sigma_i^4}\frac{\partial\sigma_i^2}{\partial\mathbf{x}_t}.$$

Plugging this into Eq. 13 yields

$$\psi_1(\mathbf{z}^{(1)}) - \psi_2(\mathbf{z}^{(2)}) = -\frac{2\alpha}{\sigma_i^4}\frac{\partial\sigma_i^2}{\partial\mathbf{x}_t}\left(\mathbf{z}_i^{(1)} - \mathbf{z}_i^{(2)}\right).$$

Global axes are those that minimize the difference in $\psi$. We can find axes that satisfy three properties:

- small variance in the population of clean image latents,
- large feature uncertainty $\sigma_i$ for our noisy image state $\mathbf{x}_t$,
- where the variance of the noisy latent $\sigma_i$ is insensitive to changes in our noisy image $\mathbf{x}_t$.

Though our analysis relies on the assumption that we are currently at a large noise regime, this is also the regime in which the guidance vector has the most influence over the sampling process, since the noise-dependent factor $\gamma_t$ in Eq. 12 —which trades off between the influence of the guidance vector and the denoiser score—scales monotonically with $t$.

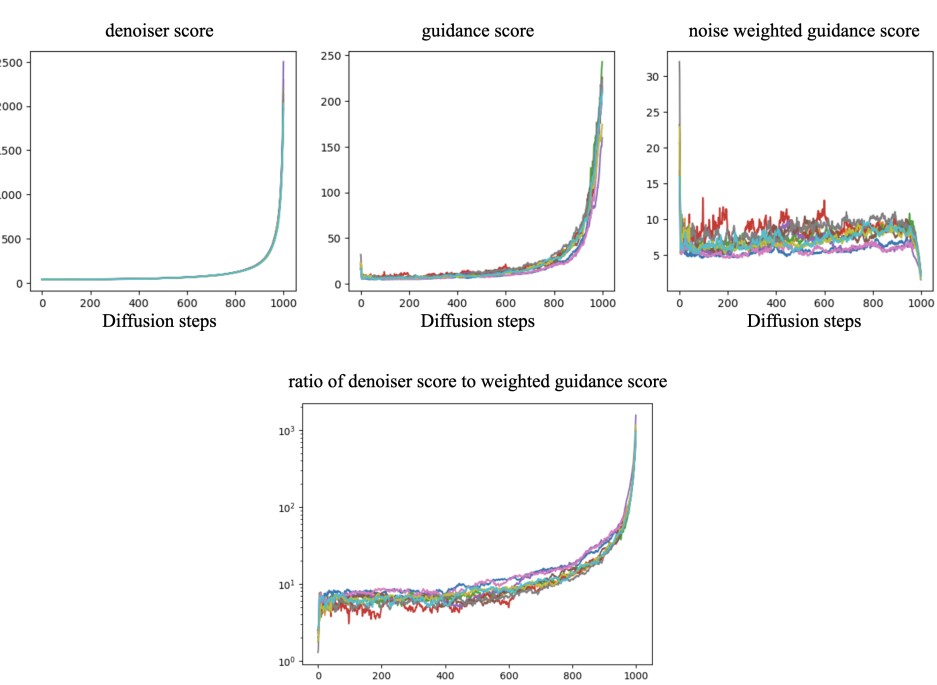

Figure 7: Comparing the norms of the denoiser score and the guidance score. Larger (rightward) denoising steps corresponds to smaller noise level. The ratio is closest to one at the start of the diffusion process, at large noise levels.

Above, we plot the ratio between the norm of the denoiser score and the norm of the weighted guidance score. A smaller ratio indicates that the guidance score has more relative strength. We see that the ratio is smallest at large noise levels, at the start of the diffusion process.

### A.9 COMPARISONS TO OTHER MODELS

In Table 3 we compare our model to the other diffusion-based representation learning models in the literature. Models with a † sign indicates that they use a separate neural network to perform

| Property | SAMI (ours) | DiffAE Preechakul et al. (2022) | DRL Mittal et al. (2023) | InfoDiff Wang et al. (2023) | PDAE Zhang et al. (2022b) | SODA Hudson et al. (2024) | DisDiff Yang et al. (2023b) | DBAE Kim et al. (2024) |
|---|---|---|---|---|---|---|---|---|
| Objective function | ELBO | MSE | MSE+L1 reg. | ELBO+MI reg. | ELBO | MSE | MSE+CE reg. | MSE |
| Guidance mechanism | Sum | Network† | Network† | Network† | Network† | Network† | Network† | Network† |
| Probabilistic latents | Yes | No | No | Yes | No | No | No | No |
| Unsupervised learning | Yes | Yes | Yes | Yes | Yes | No* | Yes | Yes |
| Using pre-trained DM | Both | No | No | No | Yes | No | No | No |
| Encodes noisy images | Yes | No | No | No | No | No | No | No |
| Enc. blind to noise level | Yes | Yes | No | Yes | Yes | Yes | Yes | Yes |
| Exact guidance score calc. | Yes | No | No | No | No | No | No | No |
| Interpretable latent axes | Yes | No | No | Yes | No | No | No | No |

Table 3: Comparison to other diffusion-based representation learning models in the literature.

guidance by approximating the gradient of the log operator. In all networks other than ours, a linear projection of the latent is used to modulate the UNet denoiser's Adaptive GroupNorm layers. The asterisk (*) in the SODA column indicates that while SODA is trained in a self-supervised manner using multiple views of the original datum, their formulation allows for unsupervised training as well. DisDiff uses additional cross entropy losses to encourage representational invariance to a subset of features. However, we find that a combinatorial representation emerges automatically from training our diffusion process to use the exact log likelihood for guidance.

# B  PROOFS

## B.1  TRAINING ON BOTH NOISY AND CLEAN IMAGES SMOOTHENS THE LATENT SPACE

We show below that training our inference network $q_\phi$ on both noisy and clean images implicitly regularizes the latent space to be smoother.

The objective (Eq. 10) has two components. The reconstruction term is equal to $\mathcal{L} = \mathbb{E}_{\mathbf{x}_0,\mathbf{x}_t,\mathbf{z}|\mathbf{x}_0} \|\mathbf{x}_0 - \hat{\mathbf{x}}_0(\mathbf{x}_t) - \beta_t^2 g_t(\mathbf{x}_t,\mathbf{z})\|^2$, where $\hat{\mathbf{x}}_0(\mathbf{x}_t)$ is estimated by a standalone MSE based denoiser and $g_t(\mathbf{x}_t,\mathbf{z}) = \nabla_{\mathbf{x}_t} \log p_\phi(\mathbf{z}|\mathbf{x}_t)$ is the guidance vector, where $q_\phi(\mathbf{z}|\mathbf{x}_t) = \mathcal{N}(\mu_\phi(\mathbf{x}_t), \mathrm{diag}(\sigma_\phi(\mathbf{x}_t)))$ is our encoder of $\mathbf{x}_t$, and the log likelihood $\log p_\phi(\mathbf{z}|\mathbf{x}_t)$ is evaluated at $\mathbf{z} \sim q_\phi(\mathbf{z}|\mathbf{x}_0)$. Let's denote the error between the clean noise and the noise estimate as the *residual* $r(\mathbf{x}_0,\mathbf{x}_t) = \mathbf{x}_0 - \hat{\mathbf{x}}_0(\mathbf{x}_t)$. In this case, the reconstruction term can be written as:

$$
\begin{aligned}
\mathcal{L} &= \mathbb{E}_{\mathbf{x}_0,\mathbf{x}_t,\mathbf{z}\sim q_\phi(\mathbf{z}|\mathbf{x}_0)} \|\mathbf{x}_0 - \hat{\mathbf{x}}_0(\mathbf{x}_t) - \beta_t^2 g_t(\mathbf{x}_t,\mathbf{z})\|^2 \\
&= \mathbb{E}_{\mathbf{x}_0,\mathbf{x}_t,\mathbf{z}\sim q_\phi(\mathbf{z}|\mathbf{x}_0)} \|r(\mathbf{x}_0,\mathbf{x}_t) - \beta_t^2 g_t(\mathbf{x}_t,\mathbf{z})\| \\
&= \mathbb{E}_{\mathbf{x}_0,\mathbf{x}_t} \|r(\mathbf{x}_0,\mathbf{x}_t)\|^2 - 2\beta_t^2 \mathbb{E}_{\mathbf{x}_0,\mathbf{x}_t,\mathbf{z}} \left[ r(\mathbf{x}_0,\mathbf{x}_t)^\top g_t(\mathbf{x}_t,\mathbf{z}) \right] + \beta_t^4 \, \mathbb{E}_{\mathbf{x}_0,\mathbf{x}_t,\mathbf{z}} \|g_t(\mathbf{x}_t,\mathbf{z})\|^2 \\
&= \mathcal{L}_1 + \mathcal{L}_2 + \mathcal{L}_3,
\end{aligned}
$$

where we have denoted each of the three terms as $\mathcal{L}_i$. Since we assume that the encoder is Gaussian, we can expand the conditional score as

$$
\log q_\phi(\mathbf{z}|\mathbf{x}_t) = -\frac{1}{2}\frac{(\mathbf{z} - \mu_\phi(\mathbf{x}_t))^2}{\sigma_\phi^2(\mathbf{x}_t)} - \log \sigma_\phi(\mathbf{x}_t) - \frac{d}{2}\log(2\pi)
$$

so now the score is

$$
\nabla_{\mathbf{x}_t} \log q_\phi(\mathbf{z}|\mathbf{x}_t) = \frac{\mathbf{z} - \mu_\phi(\mathbf{x}_t)}{\sigma_\phi^2(\mathbf{x}_t)}\nabla_{\mathbf{x}_t}\mu_\phi(\mathbf{x}_t) - \frac{(\mathbf{z} - \mu_\phi(\mathbf{x}_t))^2}{\sigma_\phi^4(\mathbf{x}_t)}\nabla_{\mathbf{x}_t}\sigma_\phi^2(\mathbf{x}_t) - \frac{\nabla_{\mathbf{x}_t}\sigma_\phi^2(\mathbf{x}_t)}{2\sigma_\phi^2(\mathbf{x}_t)}
$$

Let's make a simplifying assumption that $\sigma_\phi^2(\mathbf{x}_t)$ is not $\mathbf{x}_t$ dependent and is instead a constant value $\sigma^2$. In that case,

$$g_t(\mathbf{x}_t, \mathbf{z}) = \frac{\mathbf{z} - \mu_\phi(\mathbf{x}_t)}{\sigma^2} \nabla_{\mathbf{x}_t} \mu_\phi(\mathbf{x}_t)$$

Now let $J_\mu(\mathbf{x}_t) = \nabla_{\mathbf{x}_t} \mu_\phi(\mathbf{x}_t) \in \mathbb{R}^{d \times n}$ be the encoder Jacobian.

$$g_t(\mathbf{x}_t, \mathbf{z}) = J_\mu(\mathbf{x}_t)^\top \Sigma^{-1} (\mathbf{z} - \mu_\phi(\mathbf{x}_t))$$

where $\Sigma = \text{diag}(\sigma^2)$. Now let's assume again that $\mathbf{x}_t = \mathbf{x}_0 + \beta_t \epsilon$. We can expand $g(\mathbf{x}_t, \mathbf{z})$:

$$g_t(\mathbf{x}_t, \mathbf{z}) = g_t(\mathbf{x}_0, \mathbf{z}) + \beta_t \frac{\partial g_t}{\partial \mathbf{x}_t}\bigg|_{\mathbf{x}_t = \mathbf{x}_0} \epsilon + \frac{\beta_t^2}{2} \epsilon^\top \frac{\partial^2 g_t}{\partial \mathbf{x}_t^2}\bigg|_{\mathbf{x}_t = \mathbf{x}_0} \epsilon + \mathcal{O}(\beta_t^3)$$

Notation: if we evaluate the Jacobian at $\mathbf{x}_0$, we will denote it as $J_\mu(\mathbf{x}_0)$. Otherwise, we will use $J_\mu = J_\mu(\mathbf{x}_t)$. The zeroth order derivative:

$$g_t(\mathbf{x}_0, \mathbf{z}) = J_\mu(\mathbf{x}_0)^\top \Sigma^{-1} (\mathbf{z} - \mu_\phi(\mathbf{x}_0))$$

We can write this in index form:

$$g_t^{(k)}(\mathbf{x}_t, \mathbf{z}) = \sum_j \frac{\mathbf{z}_j - \mu_{\phi,k}(\mathbf{x}_0)}{\sigma_j^2} \frac{\partial \mu_{\phi,j}(\mathbf{x}_0)}{\partial \mathbf{x}_{t,k}}$$

Now, taking the derivative of $g_t^{(k)}(\mathbf{x}_t, \mathbf{z})$ with respect to $\mathbf{x}_{t,i}$, we get

$$\frac{\partial g_t^{(k)}}{\partial \mathbf{x}_{t,i}} = \sum_j \frac{1}{\sigma^2} \left[ -\frac{\partial \mu_{\phi,j}}{\partial \mathbf{x}_{t,i}} \frac{\partial \mu_{\phi,j}}{\partial \mathbf{x}_{t,k}} + (\mathbf{z}_j - \mu_{\phi,j}) \frac{\partial^2 \mu_{\phi,j}}{\partial \mathbf{x}_{t,i} \partial \mathbf{x}_{t,k}} \right]$$

which in tensor form is

$$\frac{\partial g_t}{\partial \mathbf{x}_t} = -J_\mu^\top \Sigma^{-1} J_\mu + \Sigma^{-1} \sum_j (\mathbf{z}_j - \mu_{\phi,j}) H_j$$

where $H_j$ is the Hessian matrix of $\mu_{\phi,j}$ with entries

$$H_j = \left[ \frac{\partial^2 \mu_{\phi,j}}{\partial \mathbf{x}_{t,i} \partial \mathbf{x}_{t,k}} \right]_{ik}$$

so we get a Hessian weighted by the encoder residual $(\mathbf{z}_j - \mu_{\phi,j})$. Putting these derivatives back in, we get:

$$g_t(\mathbf{x}_t, \mathbf{z}) = J_\mu(\mathbf{x}_0)^\top \Sigma^{-1} (\mathbf{z} - \mu_\phi(\mathbf{x}_0)) + \beta_t \left( -J_\mu^\top \Sigma^{-1} J_\mu + \Sigma^{-1} \sum_j (\mathbf{z}_j - \mu_{\phi,j}) H_j \right) + \mathcal{O}(\beta_t^2).$$

Since the guidance score $g_t(\mathbf{x}_t, \mathbf{z})$ is evaluated at the clean image latent sample, i.e. $\mathbf{z} \sim \mathcal{N}(\mu_\phi(\mathbf{x}_0), \Sigma)$, we take the expectation over the distributions $q_\phi(\mathbf{z}|\mathbf{x}_0)$:

$$\mathbb{E}_{\mathbf{z}|\mathbf{x}_0} [\mathbf{z} - \mu_\phi(\mathbf{x}_t)] = \mu_\phi(\mathbf{x}_0) - \mu_\phi(\mathbf{x}_t)$$

At $\mathbf{x}_t = \mathbf{x}_0$, the means are equal, so $\mathbb{E}_{\mathbf{z}|\mathbf{x}_0}[\mathbf{z} - \mu_\phi(\mathbf{x}_t)] = 0$. And at $\mathbf{x}_t = \mathbf{x}_0 + \beta_t \epsilon$,

$$\mathbb{E}_{\mathbf{z}|\mathbf{x}_0} [\mathbf{z} - \mu_\phi(\mathbf{x}_0 + \beta_t \epsilon)] \approx \mu_\phi(\mathbf{x}_0) - \mu_\phi(\mathbf{x}_0) - \beta_t J_\mu \epsilon = -\beta_t J_\mu \epsilon$$

This means that the expected score is

$$\mathbb{E}_{\mathbf{z}|\mathbf{x}_0} [g_t(\mathbf{x}_t, \mathbf{z})] = J_\mu^\top \Sigma^{-1} \mathbb{E}_{\mathbf{z}|\mathbf{x}_0} [(\mathbf{z} - \mu_\phi(\mathbf{x}_t))] \approx -J_\mu^\top \Sigma^{-1} \beta_t J_\mu \epsilon$$

Now we want to expand the Jacobian $J_\mu(\mathbf{x}_t)$ around the point $\mathbf{x}_0$:

$$J_\mu(\mathbf{x}_t) = J_\mu(\mathbf{x}_0) + \beta_t \sum_j \epsilon_j H_j + \mathcal{O}(\beta_t^2)$$

where we use the expression for $H_j$ that we used above. Plugging this into the expectation, we get

$$\mathbb{E}_{\mathbf{z}|\mathbf{x}_0}\left[g_t(\mathbf{x}_t, \mathbf{z})\right]$$

$$= -\beta_t \left(J_\mu(\mathbf{x}_0) + \beta_t \sum_j \epsilon_j H_j\right)^\top \Sigma^{-1} \left(J_\mu(\mathbf{x}_0) + \beta_t \sum_j \epsilon_j H_j\right)$$

$$= -\beta_t J_\mu(\mathbf{x}_0)^\top \Sigma^{-1} J_\mu(\mathbf{x}_0) + 2\beta_t^2 \left(J_\mu(\mathbf{x}_0)^\top \Sigma^{-1} \sum_j \epsilon_j H_j\right) - \beta_t^3 \sum_{j,k} \epsilon_j \epsilon_k H_j^\top \Sigma^{-1} H_k$$

$$= -\beta_t J_\mu(\mathbf{x}_0)^\top \Sigma^{-1} J_\mu(\mathbf{x}_0) + \mathcal{O}(\beta_t^2).$$

Now we will use these identities to evaluate $\mathcal{L}_2$ and $\mathcal{L}_3$.

**Expanding loss term 2**  The second term in the loss is given by

$$\mathcal{L}_2 = -2\beta_t^2 \, \mathbb{E}_{\mathbf{x}_0, \mathbf{x}_t, \mathbf{z}|\mathbf{x}_0}\left[r(\mathbf{x}_0, \mathbf{x}_t)^\top g_t(\mathbf{x}_t, \mathbf{z})\right].$$

We can use the results above for $\mathbb{E}_{\mathbf{z}|\mathbf{x}_0}\left[g_t(\mathbf{x}_t, \mathbf{z})\right] \approx -\beta_t J_\mu(\mathbf{x}_0)^\top \Sigma^{-1} J_\mu(\mathbf{x}_0)\epsilon$ to rewrite this as

$$\mathcal{L}_2 = -2\beta_t^2 \mathbb{E}_{\mathbf{x}_0, \epsilon}\left[-\beta_t r(\mathbf{x}_0, \mathbf{x}_t)^\top J_\mu(\mathbf{x}_0)^\top \Sigma^{-1} J_\mu(\mathbf{x}_0)\epsilon\right]$$

Remember that $r(\mathbf{x}_0, \mathbf{x}_t) = \mathbf{x}_0 - \hat{\mathbf{x}}_0(\mathbf{x}_0 + \beta_t\epsilon)$. If $\hat{\mathbf{x}}_0$ is a good denoiser,

$$\hat{\mathbf{x}}_0(\mathbf{x}_0 + \beta_t\epsilon) \approx \mathbb{E}\left[\mathbf{x}_0|\mathbf{x}_t\right] = \mathbf{x}_0 + \beta_t^2 \nabla_{\mathbf{x}_t} \log p(\mathbf{x}_t)$$

So

$$r(\mathbf{x}_0, \mathbf{x}_t) \approx \mathbf{x}_0 - \mathbf{x}_0 - \beta_t^2 \nabla_{\mathbf{x}_t} \log p(\mathbf{x}_t) = -\beta_t^2 \nabla_{\mathbf{x}_t} \log p(\mathbf{x}_t)$$

We can plug this back into $\mathcal{L}_2$ to get:

$$\mathcal{L}_2 = 2\beta_t^6 \mathbb{E}_{\mathbf{x}_0, \epsilon}\left[\nabla_{\mathbf{x}_t} \log p(\mathbf{x}_t) J_\mu(\mathbf{x}_0)^\top \Sigma^{-1} J_\mu(\mathbf{x}_0)\epsilon\right]$$

When we take the expectation over $\epsilon$, we find that

$$\mathcal{L}_2 = c \cdot \mathbb{E}_\epsilon\left[\epsilon\right] = 0,$$

so this entire term disappears at the leading order, only contributing at order $\mathcal{O}(\beta_t^3)$ or higher. However, we can show that this term disappears fully, even at higher orders. If $\hat{\mathbf{x}}_0(\mathbf{x}_t)$ is an optimal MSE denoiser, it computes the conditional expectation. One property of MMSE estimators is their estimation error $r(\mathbf{x}_0, \mathbf{x}_t)$ are orthogonal to any function of the observation $\mathbf{x}_t$. Since the guidance score $g_t(\mathbf{x}_t, \mathbf{z})$ is a function of $\mathbf{x}_t$, after marginalizing out the contribution of $\mathbf{z}$, this means that

$$\mathbb{E}_{\mathbf{x}_0, \mathbf{x}_t}\left[r(\mathbf{x}_0, \mathbf{x}_t)^\top \mathbb{E}_{\mathbf{z}|\mathbf{x}_0}\left[g_t(\mathbf{x}_t, \mathbf{z})\right]\right] = 0.$$

**Expanding loss term 3**  The third loss term is $\mathcal{L}_3 = \beta^4 \, \mathbb{E}_{\mathbf{x}_0, \mathbf{x}_t, \mathbf{z}}\|g_t(\mathbf{x}_t, \mathbf{z})\|^2$. From the bias-variance decomposition, we have

$$\mathcal{L}_3 = \beta_t^4 \, \mathbb{E}_{\mathbf{x}_0, \mathbf{x}_t, \mathbf{z}|\mathbf{x}_0}\left[\|g_t(\mathbf{x}_t, \mathbf{z})\|^2\right] = \beta_t^4 \, \mathbb{E}_{\mathbf{x}_0, \mathbf{x}_t}\left[\|\mathbb{E}_{\mathbf{z}|\mathbf{x}_0}\left[g_t(\mathbf{x}_t, \mathbf{z})\right]\|^2 + \mathrm{tr}\left(\mathrm{var}_{\mathbf{z}|\mathbf{x}_0}\left[g_t(\mathbf{x}_t, \mathbf{z})\right]\right)\right]$$

For the first (bias) term, we can use the identity for the expected guidance score:

$$\beta_t^4 \, \mathbb{E}_{\mathbf{x}_0, \mathbf{x}_t}\left[\|\mathbb{E}_{\mathbf{z}|\mathbf{x}_0}\left[g_t(\mathbf{x}_t, \mathbf{z})\right]\|^2\right] \approx \beta_t^6 \, \mathbb{E}_{\mathbf{x}_0, \epsilon}\left[\|J_\mu(\mathbf{x}_0)^\top \Sigma^{-1} J_\mu(\mathbf{x}_0)\epsilon\|^2\right]$$

where we have substituted the expectation over $\mathbf{x}_t$ with an functionally equivalent expectation over the noise $\epsilon$. For Gaussian $\epsilon$, we can use the identity

$$\mathbb{E}_\epsilon\left[\|\epsilon^\top A\|^2\right] = \mathrm{tr}\left(A^\top A\right) = \|A\|_F^2$$

So now

$$\beta_t^6 \mathbb{E}_{\mathbf{x}_0, \epsilon}\left[\|\epsilon^\top J_\mu(\mathbf{x}_0)^\top \Sigma^{-1} J_\mu(\mathbf{x}_0)\|^2\right] = \beta_t^6 \, \mathbb{E}_{\mathbf{x}_0}\|J_\mu(\mathbf{x}_0)^\top \Sigma^{-1} J_\mu(\mathbf{x}_0)\|_F^2$$

The second (variance) term is dependent on the variance of the score:

$$\mathrm{var}_{\mathbf{z}|\mathbf{x}_0}\left[g_t(\mathbf{x}_t,\mathbf{z})\right] = J_\mu(\mathbf{x}_t)^\top \Sigma^{-1} \mathrm{var}\left[\mathbf{z} - \mu_\phi(\mathbf{x}_0)\right]\Sigma^{-1}J_\mu(\mathbf{x}_t)$$
$$= J_\mu(\mathbf{x}_t)^\top \Sigma^{-1}\Sigma\Sigma^{-1}J_\mu(\mathbf{x}_t)$$
$$= J_\mu(\mathbf{x}_t)^\top \Sigma^{-1}J_\mu(\mathbf{x}_t)$$

Taking the trace of this gives us

$$\mathrm{tr}\left(\mathrm{var}_{\mathbf{z}|\mathbf{x}_0}\left[g_t(\mathbf{x}_t,\mathbf{z})\right]\right) = \mathrm{tr}\left(J_\mu(\mathbf{x}_t)^\top \Sigma^{-1}J_\mu(\mathbf{x}_t)\right) = \|J_\mu(\mathbf{x}_t)\Sigma^{-1/2}\|_F^2$$

If we plug the Taylor expansion of $J_\mu(\mathbf{x}_t)$ (above) into this equation, we get

$$\mathrm{var}_{\mathbf{z}|\mathbf{x}_0}\left[g_t(\mathbf{x}_t,\mathbf{z})\right] = J_\mu(\mathbf{x}_t)^\top \Sigma^{-1}J_\mu(\mathbf{x}_t)$$

$$= \left(J_\mu(\mathbf{x}_0) + \beta_t\sum_j \epsilon_j H_j\right)^\top \Sigma^{-1}\left(J_\mu(\mathbf{x}_0) + \beta_t\sum_j \epsilon_j H_j\right)$$

$$= J_\mu(\mathbf{x}_0)^\top \Sigma^{-1}J_\mu(\mathbf{x}_0) + \beta_t\sum_j \epsilon_j\left(J_\mu(\mathbf{x}_0)^\top \Sigma^{-1}H_j + H_j^\top \Sigma^{-1}J_\mu(\mathbf{x}_0)\right)$$

$$+ \beta_t^2\sum_{j,k}\epsilon_j\epsilon_k H_j^\top \Sigma^{-1}H_k + \mathcal{O}(\beta_t^3)$$

Taking the trace of this quantity up to second order terms, we get

$$\mathrm{tr}\left(\mathrm{var}_{\mathbf{z}|\mathbf{x}_0}\left[g_t(\mathbf{x}_t,\mathbf{z})\right]\right) = \|J_\mu(\mathbf{x}_t)\Sigma^{-1/2}\|_F^2$$

$$= \|J_\mu(\mathbf{x}_0)\Sigma^{-1/2}\|_F^2 + 2\beta_t\sum_j \epsilon_j\,\mathrm{tr}\left(J_\mu(\mathbf{x}_0)^\top \Sigma^{-1}H_j\right)$$

$$+ \beta_t^2\sum_{j,k}\epsilon_j\epsilon_k\,\mathrm{tr}\left(H_j^\top \Sigma^{-1}H_k\right)$$

This trace term is under expectation of $\mathbf{x}_0$ and $\epsilon$. The expectation over $\epsilon$ sets the $\beta_t$ term to 0, while the $\beta_t^2$ term becomes

$$\mathbb{E}_\epsilon\left[\beta_t^2\sum_{j,k}\epsilon_j\epsilon_k\,\mathrm{tr}\left(H_j^\top \Sigma^{-1}H_k\right)\right] = \beta_t^2\sum_j \mathrm{tr}\left(H_j^\top \Sigma^{-1}H_j\right)$$

since $\mathbb{E}_\epsilon\left[\epsilon_j\epsilon_k\right] = \mathrm{Id}_{jk} = \delta_{jk}$, i.e. 1 when $j = k$ and 0 otherwise. The first equality holds because the mean is 0 and the second equality holds because the covariance is an identity matrix. Using the identity $\mathrm{tr}(A^\top A) = \sum_{j,k} A_{jk}^2 = \sum_j \|A\|_F^2$, we can simplify this term:

$$\sum_j \mathrm{tr}\left(H_j^\top \Sigma^{-1}H_j\right) = \sum_j \frac{1}{\sigma_j^2}\mathrm{tr}\left(H_j^\top H_j\right) = \sum_j \frac{1}{\sigma_j^2}\|H_j\|_F^2\,.$$

So we are left with

$$\mathbb{E}_\epsilon\left[\mathrm{tr}\left(\mathrm{var}_{\mathbf{z}|\mathbf{x}_0}\left[g_t(\mathbf{x}_t,\mathbf{z})\right]\right)\right] = \|J_\mu(\mathbf{x}_0)\Sigma^{-1/2}\|_F^2 + \beta_t^2\sum_j \frac{1}{\sigma_j^2}\|H_j\|_F^2\,.$$

Putting together the bias and variance terms gives us:

$$\mathcal{L}_3 = \beta_t^4\,\mathbb{E}_{\mathbf{x}_0}\left\|J_\mu(\mathbf{x}_0)\Sigma^{-1/2}\right\|_F^2 + \beta_t^6\,\mathbb{E}_{\mathbf{x}_0}\left[\sum_j \frac{1}{\sigma_j^2}\|H_j\|_F^2\right] + \beta_t^6\,\mathbb{E}_{\mathbf{x}_0}\left\|J_\mu(\mathbf{x}_0)^\top \Sigma^{-1}J_\mu(\mathbf{x}_0)\right\|_F^2$$

**Combining the losses** Now, putting the second and third term back into the loss, we get

$$\mathcal{L} = \mathbb{E}_{\mathbf{x}_0,\epsilon}\|r(\mathbf{x}_0,\mathbf{x}_t)\|^2 + \beta_t^4\,\mathbb{E}_{\mathbf{x}_0}\left\|J_\mu(\mathbf{x}_0)\Sigma^{-1/2}\right\|_F^2$$

$$+ \beta_t^6\,\mathbb{E}_{\mathbf{x}_0}\left[\sum_j \frac{1}{\sigma_j^2}\|H_j\|_F^2\right] + \beta_t^6\,\mathbb{E}_{\mathbf{x}_0}\left\|J_\mu(\mathbf{x}_0)^\top \Sigma^{-1}J_\mu(\mathbf{x}_0)\right\|_F^2$$

plus higher orders. How should we interpret this loss?

The order $\beta_t^4$ term regularizes the Jacobian by encouraging the encoder to be contractive, basically penalizing the encoder from mapping small changes in the input space to large changes in latent space. It is weighted by $\Sigma^{-1/2}$, so it enforces the more informative (high-precision) dimensions to be more stable.

The second term has order $\beta_t^6$, and this penalizes the Frobenius norm of the Hessian, weighted by the precision. This forces the manifold of the latent space to be locally flat, with more informative (high-precision) dimensions getting a stronger penalty on their curvature. This ameliorates the "latent holes" issue in VAEs where latent interpolation fails by ensuring that linear changes in image space locally correspond to linear changes in latent space.

**Extending to all noise levels**   One weakness of a proof based on Taylor expansions is that it only holds when the noise level ($\beta_t$) is small. However, the variance exploding noising process we assume ($\mathbf{x}_t = \mathbf{x}_0 + \beta_t \epsilon$) allows us to write noisy images in terms of slightly less noisy images. We are going to exploit this iterative definition to apply the above proof at all noise levels. Let's define an intermediate timestep that differs from timestep $t$ by a small value $\delta t$. Due to the Markov property of the diffusion process, we can write the iterative noising process as

$$\mathbf{x}_t = \mathbf{x}_{t-1} + \delta\mathbf{x}_t,$$

where $\delta\mathbf{x}_t \sim \mathcal{N}(0, \gamma_t^2 I)$ and we have defined the incremental noise variance as $\gamma_t^2 = \beta_t^2 - \beta_{t-1}^2$. Since $\delta t$ is small, $\gamma_t^2$ is also small. We can now apply the same steps as the original proof, but instead of expanding around $\mathbf{x}_0$, we expand around the previous noisy state $\mathbf{x}_{t-1}$. This means that the Jacobian is approximated as

$$J_\mu(\mathbf{x}_t) = J_\mu(\mathbf{x}_{t-1}) + \sum_j \delta\mathbf{x}_{t,j} H_j(\mathbf{x}_{t-1}) + \mathcal{O}(\gamma_t^2)$$

$$= J_\mu(\mathbf{x}_{t-1}) + \mathcal{H}(\mathbf{x}_{t-1}) \cdot \delta\mathbf{x}_t + \mathcal{O}(\gamma_t^2).$$

where $\mathcal{H} \cdot \delta\mathbf{x}_t$ is the Hessian tensor product with a sample of scaled white noise. Since $\gamma_t$ is small, the Taylor expansions hold regardless of the size of the total noise level $t$. Now let's plug this into the third loss term that penalizes the curvature. In this case, we get an per-noise level expression in terms of Jacobians and Hessians evaluated at $\mathbf{x}_{t-1}$:

$$\mathcal{L}_3(t) = \gamma_t^4 \, \mathbb{E}_{\mathbf{x}_{t-1}} \left\| J_\mu(\mathbf{x}_{t-1})\Sigma^{-1/2} \right\|_F^2 + \gamma_t^6 \, \mathbb{E}_{\mathbf{x}_{t-1}} \left[ \sum_j \frac{1}{\sigma_j^2} \left\| H_j(\mathbf{x}_{t-1}) \right\|_F^2 \right]$$

$$+ \gamma_t^6 \, \mathbb{E}_{\mathbf{x}_{t-1}} \left\| J_\mu(\mathbf{x}_{t-1})^\top \Sigma^{-1} J_\mu(\mathbf{x}_{t-1}) \right\|_F^2.$$

However, since we train over all noise levels with some distribution, e.g. $t \sim \mathcal{U}[0, \infty]$, the total regularization effect is given by the weighted average of all $t$ dependent terms over the entire trajectory.

We can express this expected loss in terms of $\mathbf{x}_0$ by applying the Taylor expansion to the Jacobian recursively over all noise levels, which gives us

$$J_\mu(\mathbf{x}_t) \approx J_\mu(\mathbf{x}_{t-1}) + \mathcal{H}(\mathbf{x}_{t-1}) \cdot \delta\mathbf{x}_t$$

$$= J_\mu(\mathbf{x}_{t-2}) + \mathcal{H}(\mathbf{x}_{t-2}) \cdot \delta\mathbf{x}_{t-1} + \mathcal{H}(\mathbf{x}_{t-1}) \cdot \delta\mathbf{x}_t$$

$$\dots$$

$$= J_\mu(\mathbf{x}_0) + \sum_{s=1}^t \mathcal{H}(\mathbf{x}_{s-1}) \cdot \delta\mathbf{x}_s$$

In the limit of $\delta \to 0$, this sum turns into a stochastic integral with respect to the Brownian motion $\mathbf{x}_s$ of the noisy image:

$$J_\mu(\mathbf{x}_t) = J_\mu(\mathbf{x}_0) + \int_0^t \mathcal{H}(\mathbf{x}_s) \, d\mathbf{x}_s.$$

We can now plug this into the third loss term again to express it in terms of $\mathbf{x}_0$. For notational convenience, let's set $\Sigma^{-1/2} = \text{Id}$.

$$\mathcal{L}_3(t) \propto \mathbb{E}_{\mathbf{x}_0} \left\| J_\mu(\mathbf{x}_0)\Sigma^{-1/2} \right\|_F^2$$

$$= \mathbb{E}_{\mathbf{x}_0} \left\| J_\mu(\mathbf{x}_0) + \int_0^t \mathcal{H}(\mathbf{x}_s)\, d\mathbf{x}_s \right\|_F^2.$$

We can expand the squared norm term using the identity $\|A + B\|^2 = \|A\|^2 + \|B\|^2 + 2\langle A, B\rangle$, which gives us

$$\mathcal{L}_3(t) = \mathbb{E}_{\mathbf{x}_0} \|J_\mu(\mathbf{x}_0)\|_F^2 + \mathbb{E}_{\mathbf{x}_0} \left\| \int_0^t \mathcal{H}(\mathbf{x}_s)d\mathbf{x}_s \right\|_F^2 + 2\mathbb{E}_{\mathbf{x}_0} \left\langle J_\mu(\mathbf{x}_0), \int_0^t \mathcal{H}(\mathbf{x}_s)d\mathbf{x}_s \right\rangle.$$

The first term is constant over time since it is not $t$ dependent. The cross product term goes to $0$ because the expectation of the martingale is $0$. For the second term, we can use the Itô isometry, which allows us to rewrite the expected squared integral as the expected integral of the norm of the Hessians.

$$\mathbb{E}_{\mathbf{x}_0} \left\| \int_0^t \mathcal{H}(\mathbf{x}_s)d\mathbf{x}_s \right\|_F^2 = \mathbb{E}_{\mathbf{x}_0} \left[ \int_0^{\beta_t} \|\mathcal{H}(\mathbf{x}_s)\|_F^2 \, d\beta_s^2 \right].$$

If we consider all possible noise levels, the total expected loss becomes

$$\mathbb{E}_{t\sim[0,\infty]} \left[\mathcal{L}_3(t)\right] = \mathbb{E}_{\mathbf{x}_0,t} \left[ \|J_\mu(\mathbf{x}_0)\|_F^2 + \int_0^\infty \|\mathcal{H}(\mathbf{x}_s)\|_F^2 d\beta_s^2 \right].$$

**Conclusion**   By expressing the total loss in terms of the clean image, we see that we are implicitly training the encoder to minimize the integral of the Hessian, weighted by the noise level $\beta_t^2$. This essentially encourages the network to smooth the curvature along the entire denoising *trajectory*.

### B.2   Disentanglement of hierarchical features

Here we prove that a disentangled representation of hierarchical features emerges as a natural consequence of using a diffusion-based decoder with additive score-based guidance that is optimized to minimize reconstruction error and a KL divergence between a diagonal covariance posterior and an isotropic Gaussian prior. This *disentangled hierarchical representation* is one that assigns each of the dimensions in the latent space a unique feature from a hierarchy of ground truth semantic features.

**Assumption 1: hierarchy of semantic features.**   Let us assume that the images $\mathbf{x}_0 \in \mathbb{R}^D$ are formed from $K$ independent ground-truth *semantic features* $\mathbf{y} = \{y_k\}_{k=1}^K$ via a (potentially non-linear) injective, differentiable function $f$:

$$\mathbf{x}_0 = f(\mathbf{y}) = f(y_1, y_2, \ldots, y_k).$$

We assume these semantic features $\{y_k\}_{k=1}^K$ are structured hierarchically, i.e. they decay monotonically as a function of noise level $t$ and coarseness $k$. More specifically, if noisy images at noise level $t$ are defined as $\mathbf{x}_t = \sqrt{\bar{\alpha}_t}\mathbf{x}_0 + \sqrt{1 - \bar{\alpha}_t}\,\epsilon$, where $\epsilon \sim \mathcal{N}(0, I)$, then coarser features are more informative:

$$\forall t > 0, \quad I(y_1; \mathbf{x}_t) > I(y_2; \mathbf{x}_t) > \cdots > I(y_k; \mathbf{x}_t).$$

The information about each feature in the noisy image decreases monotonically with the degree of noise, so

$$\frac{\partial I(y_k; \mathbf{x}_t)}{\partial t} < 0.$$

Since fine features decay faster with noise than coarse features, these features have a third property

$$\frac{\partial}{\partial k} \left| \frac{\partial I(y_k; \mathbf{x}_t)}{\partial t} \right| > 0.$$

Since the noise level scales monotonically with the SNR of the noisy image, this means that for each semantic factor $y_k$ we can assign a *characteristic noise level* $\tau_k$, which is the highest noise level (ie. the timestep in the diffusion process) at which they are still informative. When $t > \tau_k$, $I(y_k; \mathbf{x}_t) \approx 0$.

**Assumption 2: factorized posterior**   Following the standard VAE framework, we constrain the variational family of the encoder to be a multivariate Gaussian with a diagonal covariance structure, $q_\phi(\mathbf{z}|\mathbf{x}_t) := \mathcal{N}(\mathbf{z}; \boldsymbol{\mu}_\phi(\mathbf{x}_t), \text{diag}(\boldsymbol{\sigma}_\phi^2(\mathbf{x}_t)))$. Because the covariance matrix is diagonal, the joint distribution of the latent variables factorizes into the product of marginals:

$$q_\phi(z_1, \ldots, z_D|\mathbf{x}_t) = \prod_{i=1}^{D} q_\phi(z_i|\mathbf{x}_t).$$

This factorization implies that the latent dimensions are *conditionally independent* given the observation, such that $\forall i \neq j$, $z_i \perp z_j|\mathbf{x}_t$. This also implies that $I(z_i; z_j|\mathbf{x}_t) = 0$.

**Goals of the proof**   We want to show that each of the semantic features that form our observed clean image $\mathbf{x}_0$ are assigned a specific latent variable (ie. axis/dimension) in the latent space, and this mapping is maintained as we change the noise level. To do this, we must show that two conditions are met by our model: a) at a particular noise level $t$, we have disentangled factors, and b) these latent assignments are stationary across all noise levels.

**Part I: disentanglement at a particular noise level**   The goal of SAMI is to learn a representation that best helps a denoiser recover a true image $\mathbf{x}_0$ from a noisy image $\mathbf{x}_t$ across many noise levels. The quality of the prediction is bounded by how much information about $\mathbf{x}_0$ is contained within $\mathbf{z}$ from observation $\mathbf{x}_t$. In information theoretic terms, this means the encoder want to maximize $I(\mathbf{x}_0; \mathbf{z}|\mathbf{x}_t)$ at each noise level $t$. Since $\mathbf{x}_0$ can be decomposed into its generative factors, we can express $I(\mathbf{x}_0; \mathbf{z}|\mathbf{x}_t)$ as

$$I(\mathbf{x}_0; \mathbf{z}|\mathbf{x}_t) = I(y_1, y_2, \ldots, y_K; \mathbf{z}|\mathbf{x}_t)$$
$$= \sum_{k=1}^{K} I(y_k; \mathbf{z}|\mathbf{x}_t, y_1, y_2, \ldots, y_{k-1}) + I(\mathbf{x}_0; \mathbf{z}|\mathbf{x}_t, \bar{y})$$
$$= \sum_{k=1}^{K} I(y_k; \mathbf{z}|\mathbf{x}_t) + I(\mathbf{x}_0; \mathbf{z}|\mathbf{x}_t, \bar{y})$$

where we have used chain rule in the first line and the conditional independence of the factors in the second. $\bar{y}$ are the factors not captured by the generative model, but for a good generative model we assume that this information quantity is small, so $I(\mathbf{x}_0; \mathbf{z}|\mathbf{x}_t, \bar{y}) \approx 0$. This means that only a subset of the terms in the sum above will be considered.

Now let's consider a feature $y_k$ with a characteristic noise level $\tau_k$. For $t < \tau_k$, feature $y_k$ is not detectable from $\mathbf{x}_t$, so $I(y_k; \mathbf{z}|\mathbf{x}_t) \approx 0$. For $t \geq \tau_k$, the factor is present in $\mathbf{x}_t$, so $I(y_k; \mathbf{z}|\mathbf{x}_t) > 0$.

**Proof by contradiction**   Suppose two latents $z_i$ and $z_j$ both encode information about the same feature $y_k$, such that $I(z_i; y_k|\mathbf{x}_t) > 0$ and $I(z_j; y_k|\mathbf{x}_t) > 0$. We want to show that this sort of redundancy is at odds with one of our underlying assumptions. First, let's drop the conditioning on $\mathbf{x}_t$ since it applies to all terms equally, and add it in later. The mutual information can be expressed as

$$I(z_i, z_j; y_k) = I(z_i; y_k) + I(z_j; y_k|z_i)$$

The conditional mutual information can be written as

$$I(z_j; y_k|z_i) = I(z_j; y_k) - I(z_j; y_k; z_i)$$

where $I(z_j; y_k; z_i)$ is the mutual information of three variables, otherwise known as the interaction information. Putting this together gives us

$$I(z_i; y_k) + I(z_j; y_k) = I(z_i, z_j; y_k) + I(z_j; y_k; z_i).$$

This can be bounded in two ways. First, the total information extracted by the pair $(z_i, z_j)$ cannot exceed the entropy of the source:

$$I(z_i, z_j; y_k) \leq H(y_k).$$

Second, the interaction information is bounded by the mutual information of the latents:

$$I(z_i; y_k; z_j) \leq I(z_i; z_j).$$

This means that the above identity can be written as:

$$I(z_i; y_k) + I(z_j; y_k) \leq I(z_i; z_j) + H(y_k).$$

Re-introducing the conditioning on $\mathbf{x}_t$, we get

$$I(z_i; y_k|\mathbf{x}_t) + I(z_j; y_k|\mathbf{x}_t) \leq I(z_i; z_j|\mathbf{x}_t) + H(y_k|\mathbf{x}_t).$$

If we want the latents to maximize information about $y_k$, we must maximize the left hand side. Since $H(y_k|\mathbf{x}_t)$ is fixed, increasing the left hand side means also increasing $I(z_i; z_j|\mathbf{x}_t)$ on the right hand side. However, we are constrained by the diagonal posterior assumption, which minimizes the latent correlation $I(z_i; z_j|\mathbf{x}_t) \approx 0$. This contradiction on the behavior of $I(z_i, z_j|\mathbf{x}_t)$ means that the only way to maximize information about $y_k$ while minimizing the conditional independence constraint is if only one latent variable provides information about $y_k$. This means disentanglement (ie. assignment of a particular axis to a single latent feature) is optimal given a particular noise level.

**Part II: stationarity of latents over multiple noise levels** We now show that a single latent dimension $z_i$ must track the *same* feature $y_k$ across all noise levels $0 < t < T$, and this means allocating a single latent to a unique feature across all scales.

First, we define the semantic tangent vector $\mathbf{v}_k(\mathbf{x}_0)$ as the partial derivative of the image with respect to the $k$-th feature. This vector represents the direction in pixel space corresponding to a change in feature $y_k$:

$$\mathbf{v}_k := \frac{\partial f}{\partial y_k} \in \mathbb{R}^D$$

We assume that in a high dimensional space $\mathbb{R}^D$, distinct semantic directions are approximately orthogonal, such that $\mathbf{v}_j^\top \mathbf{v}_k \approx 0$.

We will leverage the fact that SAMI uses additive guidance in score space to make geometric arguments about optimality:

$$\nabla_{\mathbf{x}_t} \log p(\mathbf{x}_t|\mathbf{z}) = \nabla_{\mathbf{x}_t} \log p(\mathbf{x}_t) + \nabla_{\mathbf{x}_t} \log q_\phi(\mathbf{z}|\mathbf{x}_t)$$

where the guidance term $\mathbf{g}_t := \nabla_{\mathbf{x}_t} \log q_\phi(\mathbf{z}|\mathbf{x}_t)$ guides the generation process. The update at time $t$ moves $\mathbf{x}_t$ in the direction of the score of the posterior. We define the guidance vector $\mathbf{g}_{i,t}$ derived from a single latent dimension $z_i$ as:

$$\mathbf{g}_{i,t} := \nabla_{\mathbf{x}_t} \log q_\phi(z_i|\mathbf{x}_t)$$

Assuming a Gaussian encoder $q_\phi(\mathbf{z}|\mathbf{x}_t) = \mathcal{N}(\mathbf{z}; \mu_\phi(\mathbf{x}_t), \sigma^2 I)$, and making a simplifying assumption that the variance is not a function of the input $\mathbf{x}_t$, the log-likelihood is proportional to the Mahalanobis distance between the predicted latent and the target guidance $\mathbf{z}^{(0)} \sim q_\phi(\mathbf{z}|\mathbf{x}_0)$:

$$\log q_\phi(\mathbf{z}^{(0)}|\mathbf{x}_t) = -\frac{1}{2\sigma^2} \|\mathbf{z}^{(0)} - \mu_\phi(\mathbf{x}_t)\|^2 + c.$$

For a single dimension $z_i$, this becomes

$$\log q_\phi(z_i^{(0)}|\mathbf{x}_t) = -\frac{1}{2\sigma^2} (z_i^{(0)} - \mu_i(\mathbf{x}_t))^2 + c,$$

where $z_i^{(0)}$ is the target latent value derived from the clean image. Taking the gradient gives us

$$\mathbf{g}_{i,t} = \nabla_{\mathbf{x}_t} \left( -\frac{1}{2\sigma^2} (z_i^{(0)} - \mu_i(\mathbf{x}_t))^2 \right) \tag{14}$$

$$= -\frac{1}{\sigma^2} (z_i^{(0)} - \mu_i(\mathbf{x}_t)) \cdot \nabla_{\mathbf{x}_t} (z_i^{(0)} - \mu_i(\mathbf{x}_t)) \tag{15}$$

$$= \frac{1}{\sigma^2} (z_i^{(0)} - \mu_i(\mathbf{x}_t)) \cdot \nabla_{\mathbf{x}_t} \mu_i(\mathbf{x}_t). \tag{16}$$

Let scalar $r_t = (z_i^{(0)} - \mu_i(\mathbf{x}_t))/\sigma^2$ be the magnitude of the error, and $J_i(\mathbf{x}_t) = \nabla_{\mathbf{x}_t} \mu_i(\mathbf{x}_t)$ be the gradient of the encoder output with respect to the input pixels. The Jacobian measures the sensitivity of the mean encoder to changes in the noisy image. We can therefore express the guidance vector as:

$$\mathbf{g}_{i,t} = r_t \cdot J_i(\mathbf{x}_t)$$

The goal is to show two things: in the high noise regime, a latent must encode a coarse feature to be useful, and in the low noise regime, switching the mapping of the latent from a coarse feature to a fine scale feature results in ineffective guidance.

**High noise regime.** Let the noisy image be $\mathbf{x}_t(\mathbf{y}) = \sqrt{\bar{\alpha}_t} \, f(\mathbf{y}) + \sqrt{1 - \bar{\alpha}_t} \epsilon$. The sensitivity of the encoder output $\mu_i$ to the ground truth feature $y_k$ can be decomposed via the chain rule as:

$$\frac{\partial \mu_i}{\partial y_k} = \left( \frac{\partial \mu_i}{\partial \mathbf{x}_t} \right)^\top \left( \frac{\partial \mathbf{x}_t}{\partial y_k} \right) = J_i(\mathbf{x}_t)^\top \cdot \left( \sqrt{\bar{\alpha}_t} \mathbf{v}_k \right)$$

We define the characteristic noise level $\tau_k$ such that for $t < \tau_k$, the mutual information $I(\mu_i(\mathbf{x}_t); y_k) = 0$. In this case, the estimator $\mu_i(\mathbf{x}_t)$ is statistically independent of $y_k$. Therefore, the expected gradient of the estimator with respect to the feature must be zero:

$$\mathbb{E}_\epsilon \left[ \frac{\partial \mu_i(\mathbf{x}_t)}{\partial y_k} \right] = 0 \implies \mathbb{E}_\epsilon [J_i(\mathbf{x}_t)^\top \mathbf{v}_k] = 0$$

This means at high noise levels, the encoder sensitivity $J_i(\mathbf{x}_t)$ is orthogonal to the feature direction $\mathbf{v}_k$ and it cannot provide guidance to recover $y_k$. This proves that latent dimensions active at $t$ must encode coarse features with characteristic noise levels $\tau_k > t$.

**Low noise regime.** Let's consider the reverse diffusion process as an integration of score updates over time $s$ from $T$ to $0$. The total displacement provided by latent $z_i$ is:

$$\Delta \mathbf{x} = \int_0^T \mathbf{g}_{i,t} \, ds = \int_0^T r_s J_i(\mathbf{x}_s) \, ds$$

If we want to recover a specific feature $y_k$, we want to maximize the projection of this displacement to the corresponding feature direction $\mathbf{v}_k$.

Let us consider the case where latent $z_i$ switches its mapping from encoding a coarse feature $y_{\text{coarse}}$ when $s > \tau_{\text{fine}}$ to encoding a fine feature $y_{\text{fine}}$ when $s < \tau_{\text{fine}}$. In the first scenario, the latent selects for a coarse feature, so the Jacobian $J_i(\mathbf{x}_s) \propto \mathbf{v}_{\text{coarse}}$. In the second scenario, the latent selects for a fine feature, so $J_i(\mathbf{x}_s) \propto \mathbf{v}_{\text{fine}}$. If we compute how much this latent recovers the fine feature, we compute the projection:

$$\Delta \mathbf{x}^\top \mathbf{v}_{\text{fine}} = \left( \int_0^T r_s J_i(\mathbf{x}_s) \, ds \right)^\top \mathbf{v}_{\text{fine}} \tag{17}$$

$$= \int_0^{\tau_{\text{fine}}} r_s J_i(\mathbf{x}_s)^\top \mathbf{v}_{\text{fine}} \, ds + \int_{\tau_{\text{fine}}}^T r_s J_i(\mathbf{x}_s)^\top \mathbf{v}_{\text{fine}} \, ds. \tag{18}$$

When $s < \tau_{\text{fine}}$, the Jacobian $J_i(\mathbf{x}_s)$ is aligned with $\mathbf{v}_{\text{fine}}$, so the first integrand is positive. However, when $s > \tau_{\text{fine}}$, the Jacobian $J_i(\mathbf{x}_s)$ is aligned with $\mathbf{v}_{\text{coarse}}$. Since distinct semantic feature are orthogonal, $\mathbf{v}_{\text{coarse}}^\top \mathbf{v}_{\text{fine}} \approx 0$, and the integrand is zero. Thus the guidance provided during the high noise regime contributes nothing to the recovery of fine feature.

Now let us consider an alternative case where we have separate latents $z_j, z_k$ dedicated to the two features $y_{\text{coarse}}$ and $y_{\text{fine}}$ respectively. In this case, it is clear that $z_j$ provides updates along $\mathbf{v}_{\text{coarse}}$ for $0 < s < \tau_{\text{coarse}}$. Meanwhile, $z_k$ is inactive during $\tau_{\text{fine}} < s < T$ and provides updates along $\mathbf{v}_{\text{fine}}$ for $0 < s < \tau_{\text{fine}}$.

In the "switching" hypothesis (single $z_i$), the capacity of $z_i$ during $\tau_{\text{fine}} < s < T$ is used to push along $\mathbf{v}_{\text{coarse}}$. This displacement is orthogonal to the target of the second phase $\mathbf{v}_{\text{fine}}$. Because the diffusion process is strictly additive, we cannot "transform" the displacement along $\mathbf{v}_{\text{coarse}}$ into displacement along $\mathbf{v}_{\text{fine}}$. Therefore, any gradient energy spent pushing along $\mathbf{v}_{\text{coarse}}$ is wasted with respect to the objective of minimizing error in $y_{\text{fine}}$. The switching strategy is thus less efficient than allocating separate dimensions.

To maximize the projection $\Delta \mathbf{x}^\top \mathbf{v}_k$ for all $k$, the Jacobian $J_i(\mathbf{x}_t)$ must maintain a constant direction $\mathbf{v}_k$ for the entire duration where the gradient is non-zero.

**Conclusion.** By combining these two constraints, we derive the optimal strategy for allocating latents to features:

1. To be useful at $t_{\text{high}}$, $z_i$ must encode a coarse feature $y_k$.

2. Even when $t$ becomes small, $z_i$ must *continue* to encode $y_k$.

This necessitates a disentangled, hierarchical representation where latent axes are sorted by the characteristic noise scale $\tau_k$ of the features they encode.

**Comments and empirical support**   The veracity of our proof rests on the two assumptions mentioned at the start. The second assumption is baked into our encoder architecture, so it is true by construction. For the first assumption, one piece of evidence that this is true is that the posterior variance along each axis increases with noise level (Fig. 3D), but not at the same rate. Moreover, most axes maintain their relative ordering over all noise levels. Altogether, this indicates that each axis encodes a semantic feature with a unique characteristic noise level. For CelebA, we find this to be true, with individual latent axes possessing unique semantic attributes (Fig. 3E).

## B.3   OPTIMALITY OF SNR-ADAPTIVE DECODER VARIANCE

SAMI is in some sense optimal under the analysis performed in Dai and Wipf (2018). This paper states that blurry reconstruction in VAEs stem from assumptions of a decoder of the form $p_\theta(\mathbf{x}|\mathbf{z}) := \mathcal{N}(\mu_\theta(\mathbf{z}), \gamma I)$, where $\gamma$ is typically fixed to 1. Theorem 3 and 4 from the paper state that minimization of the VAE objective and accurate estimation of the ground truth $\mathbf{x}$ can be achieved by reducing $\gamma \to 0$, ie. by reducing the variance of the Gaussian decoder. Indeed, this is very similar to what we do in SAMI: one way to interpret the conditional diffusion decoder is to view as a series of Gaussian decoders with decreasing $\gamma$.

This strategy is optimal according to their analysis: Eq. 9 demonstrates that lower $\gamma$ values more aggressively penalize the number of active dimensions in the latent space (scaling as $-\hat{r}\log\gamma$). This suggests that "in the neighborhood of optimal solutions the VAE will naturally seek to product perfect reconstructions using the fewest number of clean, low-noise latent dimensions", where the number of utilized dimensions is equivalent to the manifold dimension in the data.

However, to the best of our knowledge, the dimensionality of the natural image manifold appears to be highly dependent on the SNR (Guth et al., 2025), such that clean images effectively live in a manifold requiring the full ambient dimensionality, while highly corrupted images live on much lower dimensional manifolds. This is problematic for VAEs that seek to represent both clean and noisy images. According to Eq. 9, accurately reconstructing clean images requires a decoder with low $\gamma$, which incurs a large regularization cost per dimension. However, using the same low-$\gamma$ decoder on noisy images forces the model to treat noise as signal, preventing the pruning of uninformative latent dimensions and causing the KL term to explode.

Given a fixed dimensional latent space, the optimal solution is to modulate $\gamma$ depending on the SNR. Noisy images with low intrinsic dimensionality should use decoders with larger $\gamma$ to avoid overfitting noise and reduce the dimension penalty, while clean images should use decoders with smaller $\gamma$ to resolve high-dimensional, fine scale details. This is essentially what SAMI does: the conditional denoiser has the same functional form as a Gaussian decoder whose variance is paired to that of the observation noise. As we see empirically in Fig. 3D, the number of low-noise latent dimensions recruited is a function of the ambient dimensionality of the noisy image manifold from which $\mathbf{x}_t$ is sampled.