# OpenReview forum: "Unsupervised learning of disentangled representations via diffusion variational autoencoders"
_ICLR.cc/2026/Conference — Submitted to ICLR 2026_

### Official Review · Reviewer_qY7s · 2025-10-31

**Soundness:** 2
**Presentation:** 2
**Contribution:** 2
**Rating:** 4
**Confidence:** 4

**Summary:**

The paper presents DiVA, a diffusion variational autoencoder for disentangled representation learning by combining a diffusion model with a VAE. The training loss is formalized under a unified ELBO objective.

**Strengths:**

- A novel approach that combines VAEs with diffusion-based models.
- The new objective formalized of the ELBO.
- Good results on the selected benchmarks and datasets.

**Weaknesses:**

**Major:**
- Unorganized results and a lack of detailed reporting in several sections.
- Lack of comprehensive benchmarks on the suggested datasets.
- Some experiments are missing important implementation and protocol details.

**Minor:**
- Limited discussion of related literature on sequential disentanglement for video trajectories.

**Questions:**

1. **In results in Synthetic Disks dataset:** a. Why the authors report MSE for these results? Unconditional generations trending toward noise could artificially lower MSE; please justify this choice or include complementary metrics.
b. Why not use 2D shapes dataset (e.g., those used in β-VAE) for comparability?

2. **In results CelebA dataset:** a. Lines 321–323: what was the reduction in MSE? b. Why don’t you refer to the DBAE + TC results in Table 3 directly within this section?

3. Why the images are gray in Figure 3?

4. **Feature extraction for pre-trained diffusion models:** a. Can the authors show any additional results for this experiment? Also, please add more details about the fine-tuning protocol.
b. What are the pros and cons of using pre-trained backbones versus training from scratch?
c. Does this approach achieve results similar to joint training? By how much does it reduce training time?

5. **In results Encoding of video trajectories:**
a. There is a rich literature on sequential disentanglement that isn’t cited, such as [1] and [2].
b. Do you have any results demonstrating disentanglement on CelebA-HQ?
c. What is the motivation for using cosine similarity? How does it support the hypothesis in lines 438–440?
d. Why do you only compare against DiffAE?
e. Why the authors do not include benchmarks on a video dataset with labels for static and dynamic factors (e.g., Moving dSprites or related variants)?

**References**

[1] "Sequential Representation Learning via Static-Dynamic Conditional Disentanglement" M. Cyrille Simon et al.

[2] "Sequential Disentanglement by Extracting Static Information From a Single Sequence Element" N. Berman et al.

---

> ### Author Response · Authors · 2025-11-21
> **Weaknesses and Q1-3**
>
> **Response to weaknesses**\
> Could you be more specific in what details are lacking and how would you want the results to be organized?
> What kind of implementation and protocol details would be helpful to know? The details of the training procedure were originally given in the appendix, where there is also a link to an anonymized code repository. What exactly is missing?
>
> **Answers to questions**\
> Q1a. As mentioned in line 289, we want a measure of how much entropy (with variance as proxy) in the data is “explained away” by conditioning on the representation.  While this is an approximate measure, the variance is reduced by several orders of magnitude, from 5.6e-1 to 4e−3, indicating that the representation has a substantial effect.  We know from Fig. 2C that samples generated from the conditional diffusion process are high quality and resemble the conditioning image, so they are not tending towards noise. We have edited the main text to clarify what we are aiming to measure and our procedure.
>
> Q1b. While we could have used the 2D shapes/dSprites dataset, we chose the disks dataset (a variant of that used in Kadkhodaie et al., 2024 [1]) for its simplicity. What is needed here is exact knowledge and control over the underlying ground truth data generation process, against which to compare the learned latents (Figs. 2D-F).  We will include dSprites results as well in the next couple of days.
>
> Q2a. We calculated the reduction in variance (as mean squared distance to the empirical mean) on the CelebA test set. Conditioning reduces the variability from 6.62e-2 to 2.04e-3, indicating that the image specific representation explains away much of the variability in the data. We have included this result in the main text and have included a more detailed explanation in Appendix A.7.
>
> Q2b. We will reference Table 3 in this section.
>
> Q3. We trained on grayscale images because this offers much of the complexity present in natural images (at 64x64 resolution) without the large scale compute required for training on color CelebA.

---

> > ### Author Response · Authors · 2025-11-21
> > **Q4,5**
> >
> > Q4a. We are including more conditionally generated samples from a diffusion model trained on CelebA-HQ. We freeze the diffusion model weights and train an encoder using the same set of hyperparameters and training dataset as the original diffusion model, with the only change being that we sample the noise level non-uniformly with more higher noise levels.
> >
> > Q4b. Pros of using a pre-trained model: substantial increase in computational efficiency. The representation is also less likely to suffer from posterior collapse because the denoiser already provides the encoder with a strong training signal, especially at the start of training. Cons: matching hyperparameters to the original unconditional diffusion model can be prohibitive. In contrastm joint training allows the diffusion model and encoder to co-adapt.
> >
> > Q4c. Using a pre-trained denoiser often results in conditional samples with larger variance compared to those from jointly trained models. Since the encoder approximates the posterior as gaussian, a jointly trained denoiser adapts its outputs to compensate for this limited flexibility. On the other hand, a pre-trained denoiser might expect a correlational structure with higher-order moments, meaning that when we use a conditional Gaussian encoder we capture fewer of the features learned by the denoiser. We haven’t tried joint training on CelebA-HQ, but on CelebA the training time reduces from 21 hours to around 14 hours.
> >
> > Q5a. Sequential disentanglement focuses on models that takes time series as inputs. In contrast, our model takes static images as inputs, but still yields easy to predict over time latents at test time.  We will cite sequential disentanglement work in discussion to highlight the difference.
> >
> > Q5b. We do not have disentanglement results on CelebA-HQ for now, but we will add some to the appendix in the next few days.
> >
> > Q5c. The linear predictability, or straightness, of trajectories can be measured by the cosine similarity between temporally adjacent latents. A cosine similarity of 1 indicates that all latents form a straight line, while 0 tells marks two adjacent vectors that are orthogonal. Since the probability of two random vectors being orthogonal increases with the dimensionality of the vector space, the fact that our 512 dimensional latent space exhibits cosine similarity values close to 1 indicates trajectories being quite straight.
> >
> > Q5d. We only compared against DiffAE because, like almost all of the other diffusion-based representation learning models, it lacks a KL term in its objective that encourages disentanglement in latent space. Since linear trajectories of natural videos is a direct consequence of a disentangled latent space, we expect that other models that do not explicitly encourage disentanglement will also exhibit curvier trajectories as with DiffAE.
> >
> > Q5e. We did not train on video datasets because 1) our model operates on static images, 2) the aim is to show that predictive trajectories emerge via disentangling the static image features. There is no reason why training on video data where the temporal features are independent from the spatial features would result in the type of latent straightening that we observe.
> >
> > References
> > [1] Z Kadkhodaie, F Guth, E P Simoncelli, and S Mallat. (2024) Generalization in diffusion models arises from geometry-adaptive harmonic representation. In Int’l Conf on Learning Representations (ICLR)

---

### Official Review · Reviewer_p7dT · 2025-10-31

**Soundness:** 2
**Presentation:** 3
**Contribution:** 2
**Rating:** 2
**Confidence:** 4

**Summary:**

This paper proposes to condition diffusion models on a learned latent representation $z$. Using this new formulation, the authors derive a novel learning objective where the second term is equivalent to VAEs’ regularisation term. As in $\beta$-VAE, then add a scalar $\beta$ to this term to weight the regularisation and induce sparsity. The reconstruction and disentanglement abilities of the model are then evaluated on a synthetic disk dataset and on CelebA.

**Strengths:**

- The paper is well written and easy to follow
- The idea of learning meaningful latents with diffusion models is appealing

**Weaknesses:**

My main concern is about the disentanglement claim. The authors attempt to replicate $\beta$-VAE behaviour using a $\beta$ term in the second term of Eq. 5, but in the experimental section, disentanglement is not measured with any disentanglement metrics (e.g., MIG, DCI or any listed in [1]).  Furthermore, several datasets have been used to benchmark disentanglement (e.g., DSprites, SmallNorb, etc. See [1] or [2] for more examples) but apart from CelebA none of these are used here. We know from [1] that in VAEs, disentanglement capacity varies a lot depending on the dataset, so one would expect a more expansive evaluation when the authors state that their model can "recover ground truth factors". Especially given that this is not so clear cut for $\beta$ VAE or any VAE doing disentanglement. These models tend to induce sparsity with a PCA-like behaviour [3-6], and one can obtain disentangled representations if those PCs correspond to ground truth factors. Overall, I think this paper is interesting, but would need a significant rework of the empirical section to justify the disentanglement statement.

**Questions:**

- The proposed model disentanglement capacity should be evaluated using disentanglement metrics (see [1])
- The evaluation should be done on several disentanglement dataset (see [1-2])
-  Could the authors discuss the relationship between the proposed model and diffusion models being a special case of hierarchical markov VAE, as shown in [7]?
- Would this new formulation allow for other disentanglement techniques than $\beta$-VAE?
- I suggest the authors avoid saying that their model "recovers ground truth generative factor" as most disentanglement models cannot reliably do this (see [1])
- The name of the model is quite confusing, given the naming of these two previous works [8-9], and may need to be updated

References
=========
[1] Locatello, Francesco, et al. "Challenging common assumptions in the unsupervised learning of disentangled representations." international conference on machine learning. PMLR, 2019. (limitations of disentanglement)

[2] Gondal, M. W., Wuthrich, M., Miladinovic, D., Locatello, F., Breidt, M., Volchkov, V., ... & Bauer, S. (2019). On the transfer of inductive bias from simulation to the real world: a new disentanglement dataset. Advances in Neural Information Processing Systems, 32.

[3] Dai, B. et al. "Connections with robust PCA and the role of emergent sparsity in variational autoencoder models." Journal of Machine Learning Research 19.41 (2018): 1-42.

[4] Bin Dai, & David Wipf (2019). Diagnosing and Enhancing VAE Models. In International Conference on Learning Representations.

[5] Rolinek, M., Zietlow, D., & Martius, G. (2019). Variational autoencoders pursue pca directions (by accident). In Proceedings of the IEEE/CVF Conference on Computer Vision and Pattern Recognition (pp. 12406-12415).

[6] Bonheme, Lisa, and Marek Grzes. "Be more active! understanding the differences between mean and sampled representations of variational autoencoders." Journal of Machine Learning Research 24.324 (2023): 1-30.

[7] Luo, C. (2022). Understanding diffusion models: A unified perspective. arXiv preprint arXiv:2208.11970. (equivalence between diffusion models and HMVAEs)

[8] Ilse, Maximilian, et al. "Diva: Domain invariant variational autoencoders." Medical Imaging with Deep Learning. PMLR, 2020. (DiVA confusing name 1)

[9] Perez R. et al. (2020). Diffusion Variational Autoencoders. In Proceedings of the Twenty-Ninth International Joint Conference on Artificial Intelligence, ĲCAI-20 (pp. 2704–2710). International Joint Conferences on Artificial Intelligence Organization. (DIVA confusing name 2)

---

> ### Author Response · Authors · 2025-11-21
> **Weaknesses and Questions**
>
> **Regarding weaknesses**:
>
> We chose to measure disentanglement using the TAD metric (ref [a]), which has been used by other diffusion-based representation learning methods (e.g. DiffAE, InfoDiffusion, DisDiff, DBAE) to measure the degree of disentanglement in models trained on CelebA. We agree that additional disentanglement metrics could be useful. Since [1] indicates that all metrics except for Modularity are strongly correlated with each other on the dSprites dataset, we are now training our model on dSprites and will measure disentanglement with one or two of the established metrics from [1]. However, because our method performs disentanglement by taking advantage of hierarchical structure in natural image datasets (see below), we may not necessarily outperform established VAE-based methods on synthetic datasets that lack this structure. We will report the results in the next few days.
>
> We do not claim that the model is able to recover ground truth generative factors in general, but only in our synthetic dataset (Fig. 2, disks dataset) where we know the precise ground truth factors by construction.
>
> While references [3-6] provide useful insights into the notion of disentanglement in VAEs, we do not believe that all of these analyses necessarily extend to DiVA. Ref [1] suggests that while unsupervised disentanglement is impossible for any arbitrary generative model, it is feasible given a model containing the right inductive biases to take advantage of specific structure in the underlying generative model. Moreover, ref [5] shows that VAEs are only performing local PCA given a nonlinear encoder/decoder, and assumes a deterministic decoder, not a stochastic one as in our model. Finally, ref [4] assumes that the intrinsic dimensionality of the data distribution is fixed, which we know from Guth et al. 2025 [b] is untrue for natural images. For these reasons, we believe that these prior analyses are insufficient to capture the behavior of our model. We will add a new proof in the Appendix to show that DiVA performs disentanglement by exploiting structure in natural data distributions, and show also that DiVA is in some sense optimal under the analytical framework used by ref [4].
>
> **Answers to Questions:** \
> Q1) Addressed above: beyond measuring the TAD metric for CelebA, we will measure disentanglement using one or two metrics from [1] after training our model on the dSprites dataset, since [1] indicates that all of the metrics are strongly correlated on this dataset.
>
> Q2) Also addressed above: since our model is leveraging hierarchical structure in natural images, we do not necessarily believe that the disentanglement properties that we observe in CelebA will necessarily extend to most synthetic datasets that lack this hierarchical structure. However, we will train our model on the dSprites dataset as a way of comparing against existing VAE-based methods.
>
> Q3) Reference [7] states that — in general — diffusion models can be understood as Markovian Hierarchical Variational Autoencoders, which is exactly the part of the graphical model that we use to describe diffusion models in Fig. 1A, blue. Our model can be understood as a combination of this graphical model, which captures the density over the entire dataset, and a standard VAE encoder, which learns a low dimensional representation of the data. The representation is encouraged to be useful by using it to form a guidance vector that guides the diffusion process towards the ground truth observation.
>
> Q4) It is not clear what is meant here by another disentanglement technique, we’d say that (smartly done) diffusion gives extra power to the beta-VAE idea by taking advantage of the coarse-to-fine structure in natural images. Since diffusion models carry this information implicitly, it is possible that other methods for conditioning or extracting representations from unconditional diffusion models might support other disentanglement techniques than $\beta$-VAEs.
>
> Q5) Thanks for catching that, we will come up with a new acronym for the method.
>
> **References:**\
> [a] Yeats, Liu, Womble, Li. (2022) NashAE: Disentangling Representations Through Adversarial Covariance Minimization. ECCV 2022\
> [b] Guth, Kadkhodaie, Simoncelli. (2025) Learning normalized image densities via dual score matching. Neurips 2025

---

> > ### Comment · Reviewer_p7dT · 2025-11-26
> > **Given your answer, it would be great to emphasize the hierarchical aspect in the paper**
> >
> > Thank you for your detailed answer.
> >
> > I did miss the hierarchical aspect of the proposed method ('hierarchical' only appears in the refs of the paper). It would be great to emphasise this difference with previous disentanglement models in the paper.
> >
> > > Q1) Addressed above: beyond measuring the TAD metric for CelebA, we will measure disentanglement using one or two metrics from [1] after training our model on the dSprites dataset, since [1] indicates that all of the metrics are strongly correlated on this dataset.
> >
> > I agree that Modularity + any other (MIG, DCI, etc.) would be sufficient.
> >
> > > Q2) Also addressed above: since our model is leveraging hierarchical structure in natural images, we do not necessarily believe that the disentanglement properties that we observe in CelebA will necessarily extend to most synthetic datasets that lack this hierarchical structure. However, we will train our model on the dSprites dataset as a way of comparing against existing VAE-based methods.
> >
> > As discussed above, I did not realise, reding the paper the importance of the hierarchical aspect (this should be emphasised). Feel free to use another dataset such as the MPI3D dataset from Locatello et al. [1], a disentanglement dataset composed of natural images, if it fits better with this aspect.
> >
> > > Q3) Reference [7] states that — in general — diffusion models can be understood as Markovian Hierarchical Variational Autoencoders, which is exactly the part of the graphical model that we use to describe diffusion models in Fig. 1A, blue. Our model can be understood as a combination of this graphical model, which captures the density over the entire dataset, and a standard VAE encoder, which learns a low dimensional representation of the data. The representation is encouraged to be useful by using it to form a guidance vector that guides the diffusion process towards the ground truth observation.
> >
> > Maybe you could use this to emphasise the hierarchical aspect of the proposed disentanglement method?
> >
> > > Q4) It is not clear what is meant here by another disentanglement technique, we’d say that (smartly done) diffusion gives extra power to the beta-VAE idea by taking advantage of the coarse-to-fine structure in natural images. Since diffusion models carry this information implicitly, it is possible that other methods for conditioning or extracting representations from unconditional diffusion models might support other disentanglement techniques than $\beta$-VAEs.
> >
> > Sorry if I was unclear. What I meant is that there are a bunch of other disentanglement VAEs, such as $\beta$-TC VAE, DIP VAE, etc., and they generally change the ELBO slightly to encourage sparsity. Could your proposed model extend to these learning objectives as well or is it onlyh possible to add a $\beta$  term as in $\beta$ VAE?
> >
> > > We do not claim that the model is able to recover ground truth generative factors in general, but only in our synthetic dataset (Fig. 2, disks dataset) where we know the precise ground truth factors by construction.
> >
> > My comment was specifically about the claim in the abstract ``it recovers ground truth generative
> > factors in synthetic datasets'' which, for me, reads like ``it can recover ground truth generative factors from any synthetic dataset', not like ``it can recover ground truth generative factors from our specific synthetic dataset''. I suggest rephrasing this part to avoid misunderstanding.
> >
> > For now, I leave my score as it is given that the paper has not been updated. Let me know when a new version is available and I will update my score accordingly.
> >
> > [1] Locatello, Francesco, et al. "A sober look at the unsupervised learning of disentangled representations and their evaluation." Journal of Machine Learning Research 21.209 (2020): 1-62.

---

### Official Review · Reviewer_12xU · 2025-10-31

**Soundness:** 2
**Presentation:** 3
**Contribution:** 2
**Rating:** 2
**Confidence:** 4

**Summary:**

This work basically propose to learn a latent variable model along with the standard diffusion objective. By some algebra, the overall loss can be written in form of a guidance term and can be jointly trained with the unconditional score using denoising score matching plus some KL penalty on gaussian prior. It is demonstrated that the learned latent model can encode disentangled representation of the data distribution.

**Strengths:**

Combining VAE loss with standard denoising score matching to learn meaningful representation is interesting.

**Weaknesses:**

1. The proposed method is conceptually and methodologically similar to the DiffAE paper. It is unclear why DiVA performs better. It is argued this is because DiVA minimizes the exact ELBO, which is not true. In the algorithm, the weights $\lambda_t$ is not included, implying DiVA only minimizes ELBO approximately. My question is, what is the unique advantage of DIVA?

2. The trained model has to take a clean image $x_0$ as conditional input, which limits its generation capability. In figure 3 B, I would say when you condition on $x_0$, the generated images seem to be identical to it, with only minor difference. This is a sign of overfitting. Will DiVA generate high quality image when a clean image is not available? How do you calculate the FID in Table 3? Do you generate each image by input a clean image, or are the images generated unconditionally?

3. Can you provide more theoretical analysis on objective (10)? For example, what would be the optimal $q_{\phi}$ that minimizes this loss? Can you come up with some more in depth characterization of $q_{\phi}$'s property? Currently, the objective makes sense intuitively, but is kind of superficial in my opinion, as there is no theoretical guarantee that $q_{\phi}$ can capture disentangled representations. If it indeed does, why? Does it work well consistently on different dataset, or it only works on simple dataset like faces and disks?

4. Experiments are limited to simple dataset such as disks and faces. Please perform experiments on ImageNet to fully demonstrate the strength of your approach. I am not convinced if only experiments on faces and disks are provided, as nowadays, these datasets are considered too simple.

5. What is the current state of the art methods for learning disentangled representations besides the ones based on diffusion models? Does DiVA beat those algorithms? Is it really necessary to learn disentangled representation based on diffusion framework? If so, why? What is the unique advantage of diffusion in this context, from a rigorous theoretical perspective?

**Questions:**

See my questions above.

---

> ### Author Response · Authors · 2025-11-21
> **reply to Q1 and Q2**
>
> Q1 While DiVA is conceptually similar to DiffAE, our models differs in multiple ways.
>
> First, DiVA leverages the exact mathematical formulation of the conditional guidance process: the guided score $\nabla_{x_t}\log p(x_t|z)$ is the sum of the guidance score $g_t = \nabla_{x_t}\log p(z|x_t)$ and the unconditional score from the underlying diffusion process $\nabla_{x_t}\log p(x_t)$. Importantly, this math shows that the guidance score needs to be a function of the noisy image $x_t$. This stands in contrast with other diffusion based representation learning models (where the encoder network only takes the clean image $x_0$ as input), resulting in guidance score biases that must be explicitly corrected for (see: MPGD by He et al, 2024; TDS by Wu et al., 2023). The effect of this bias is further amplified by the fact that the relative influence of the guidance score in determining the noise present in the noisy image at time $t$ (Eq. 10) is modulated by the standard deviation of the noise, meaning that the bias introduced by using the clean image as opposed to the noisy image is largest when the relative effect of the guidance score is at its strongest. Our innovation is to provide a mathematically precise way to learn representations via the construction and use of the guidance score in the context of diffusion models (Eq. 9). Thus, while there are conceptual overlaps between our model and prior work, our solution is exact (also simpler and more elegant), and relies on Bayesian mathematical properties to introduce a more efficient and interpretable parameterization of the problem.
>
> Second, unlike DiffAE, our encoder $q_\phi$ is probabilistic, meaning that our representation $z$ has an associated uncertainty. This uncertainty aides interpretability in several important ways: not only does the uncertainty denote the model’s confidence in the identified features (Fig. 3C), but it also provides a signal that can be used to interpret the global nature and semantic meaning of individual axes (Fig 3F, G). Furthermore, the probabilistic nature of the representation allows for the KL regularization term that stems from the ELBO, which in turn encourages the model to find independent factors of variability (Fig. 3E).
>
> In many of our models the weights $\lambda_t$ in our objective are set to 1 but we have also trained models using weights that correspond to the variational lower bound, as derived by Kingma et al. 2021. We found that the weight distribution had negligeable impact on the denoising performance across all noise levels and were overshadowed by the effect of other hyperparameters such as the number of noise levels. Thus, while our ELBO is approximate, from empirical observation it is likely that the advantage of our model stems not from implementing the exact ELBO but from a combination of the probabilistic encoder and the use of the exact guidance procedure via an encoder that has access to both clean and noisy images.
>
> Q2 There seems to be a fundamental misunderstanding about the conditioned generation process that may have lead to the overfitting comment.  The conditioning image is part of a held-out test set and is never seen in the training process. Hence the visual similarity between the generated images and the conditioning image is not a sign that the model has overfit to (or has memorized) the training set. Instead, it is a sign that our model has not only learned a good representation of the semantic attributes of the image, but also possesses a good image prior that appropriately “fills in” the fine scale structure. It’s hard to say how much variability should one expect from this process, but careful inspection of the four conditionally generated images in Fig. 3B shows that each image does exhibit differences in the fine scale features.  Moreover, the model can generate high quality samples when conditioned on a noisy version instead of clean images (given that the noise level is not so large as to alter the semantics); this further argues for coarse features being specified in the conditioning and the pixel level prior filling in the details. For visual comparison, we have edited the appendix to include examples of conditioned generation starting from noisy versions of test images. More broadly, conditioned generation is a standard way for assessing unsupervised learning methods, such as VAEs and other diffusion-based representation learning methods (see DiffAE, InfoDiffusion, DisDiff). We followed the standard procedure for computing the FID in diffusion-based representation learning models, which is to conditionally generate a new image for each clean image in the test set based on a random seed, rather than via unconditional generation.

---

> ### Author Response · Authors · 2025-11-21
> **reply Q3-5**
>
> Q3 We will include a proof to show that the optimal behavior for the encoder $q_\phi$ is to learn a disentangled representation that assigns unique latent axes to different features in the dataset. In brief, we show that this is because the underlying diffusion model learns a hierarchical, coarse-to-fine representation of the data distribution by learning to denoise at different noise levels. Given the KL term that encourages factorization of the posterior, the optimal behavior is to learn a bijection that maps features at different scales to their own latent axes. We believe the reason why our model outperforms VAEs on disentanglement is because it is leveraging the multiscale $1 / f$ distribution of natural images (such as those of faces); we imagine it would also do well on other datasets that are distributed in this way.
>
> Q4) As researchers at an academic institution, we do not have the computational resources to train our model on ImageNet from scratch. We pragmatically chose to demonstrate the features of the method using the smallest datasets that still have the core featured required to make the point: the disk dataset shows that we can recover ground truth factors known a priori, and CelebA (regular and HQ) show that semantically meaningful axes emerge in natural datasets. Our main contributions are conceptual, demonstrating that there is a more mathematically precise way to learn a probabilistic, disentangled representation when compared to existing diffusion-based representation learning models.
>
> Q5) The current state of the art methods for disentanglement which do not use diffusion are VAEs (e.g. InfoVAE and $\beta$-TCVAE). We show in Appendix A7 that we comfortably beat VAE-based models as well as diffusion-based representation learning models on the CelebA dataset, as measured by the TAD metric for disentanglement. While diffusion is not strictly needed for disentanglement, it does provide some concrete advantages over VAEs, especially within our formalism: the generative model becomes richer and the recognition model is forced to consistently encode features (and uncertainty) across noise levels. This regularizes the learned representation to be smoother (thereby overcoming the “latent holes” problem in VAEs) and to be more disentangled. We will update the main text/appendix with both proofs in a few days.

---

> > ### Comment · Reviewer_12xU · 2025-11-22
> >
> > Please remind me again after the paper update is completed.

---

### Official Review · Reviewer_2F6a · 2025-11-01

**Soundness:** 3
**Presentation:** 2
**Contribution:** 2
**Rating:** 4
**Confidence:** 3

**Summary:**

The paper proposes an unsupervised representation disentanglement model, the Diffusion Variational Autoencoder (DiVA). Specifically, DiVA integrates variational autoencoders and diffusion models to enable unsupervised learning of structured and interpretable latent representations with strong factorization and semantic consistency, while maintaining high-quality generative performance. The proposed model is evaluated on both synthetic and real-world datasets and compared against several baseline methods.

**Strengths:**

1. The paper addresses the problem of learning disentangled representations for image data, which is an important and long-standing research topic.
2. The idea of combining the advantages of diffusion models and VAEs is conceptually clear and technically sound.
3. The paper provides a detailed analysis of the ELBO formulation.

**Weaknesses:**

1. The comparison between the proposed model and baseline methods, particularly diffusion-based disentanglement models, is unclear and difficult to follow. The key results in Appendix A.6–A.7 are important and should be highlighted in the main text. Moreover, no qualitative examples or visual comparisons are provided to illustrate the superiority of the proposed model over recent baselines such as InfoDiff, DisDiff, and DBAE+TC.
2. The InfoDiffusion model appears highly relevant, as it also includes ELBO analysis and mutual information regularization, but the differences between the two approaches are not systematically discussed.
3. The evaluation is limited to one synthetic and one real-world dataset. Given that multiple public datasets with ground-truth disentanglement factors (e.g., 3DShapes, dSprites, etc) and commonly used real-world datasets (e.g., CelebA, FFHQ, etc) are available, the experimental validation seems insufficient.

**Questions:**

Please see "Weaknesses"

---

> ### Author Response · Authors · 2025-11-21
> **Reply to questions**
>
> 1. We have included substantial review of alternative methods verbally in the main text and extended point to point comparisons as tables in the Appendix.  It is not obvious to us how to improve the comparison further, please clarify. We do agree that the results in appendix A.6 and A.7 are important. Due to space considerations we could not include them in the main text but we can edit the text to further emphasize them. We will provide additional visual comparisons against recent baselines in the Appendix in the next days.
>
> 2. InfoDiffusion is related to our work, as we originally recognized in the main text (pg 7 line 359), and we present key differences to many of the baseline diffusion-based representation models in a systematic manner in the Appendix (A.6).\
> \
> There are however several important differences.
>     - a) The key difference is that the loss of InfoDiff includes an additional term that explicitly aims to maximize mutual information. In contrast, our derivations show that this is unnecessary: we can show, based on work done by references [a, b, c], that a denoising-based generative model trained to minimize conditional MSE, such as that used in our model, already implicitly maximizes MI.
>
>     - b) A second key difference is about InfoDiff not taking advantage of the probabilistic nature of the problem the way we do, which endows our solution with additional beneficial properties in terms of smoothness of the representation and interpretability.
>
>     - c) Finally, unlike alternatives, we formally derive the ELBO for this particular model, which justifies not only the exact form of the loss, but also architectural choices such as the parameter sharing between recognition and conditioned generation. These are not hacks or vague intuitions, but come directly from the math.
>
> 3. The entire point of training on synthetic datasets such as dSprites is to have be able to validate learned representations against ground truth factors. The disk dataset serves the same goal. That said, we will include results on dSprites as well for completeness.\
> \
> With regards to training on commonly used real-world datasets, we trained our model on CelebA (64x64, grayscale, results shown in Fig. 3), and extracted a representation from an unconditional diffusion model pre-trained on CelebA-HQ (256x256, color, results in Fig. 4). While we agree that it would be nice to demonstrate how this model scales to even more difficult datasets such as FFHQ, it is almost impossible to train huge models within a reasonable amount of time, especially given the human and compute resources available at a typical academic institution. The goal here is to present a new model idea based on clear mathematical derivation, and the representational consequences of its associated probabilistic representation; we believe that the inclusion of CelebA-HQ already shows the generality of the method for large scale models to the point where it can motivate application-focus researchers to use it productively.
>
> **References:**\
> [a] Kong, Liu, Li, Yogatama, Ver Steeg. (2024) Interpretable Diffusion via Information Decomposition.\
> [b] Franzese, Bounoua, Michiardi. (2024) MINDE: Mutual Information Neural
> Diffusion Estimation.\
> [c] Guo, Shamai, Verdu. (2005) Mutual information and minimum mean-square error in Gaussian channels. IEEE Transactions on Information Theory.

---

### Author Response · Authors · 2025-12-03
**Summary of new theoretic and numerical results, and responses to reviewers**

We would like to thank the reviewers for their helpful feedback. We have incorporated many of your suggestions in our updated manuscript. Here we summarize the reviewer-specific changes we made to the manuscript, as well as new results that address the points of weakness identified by the reviewers.

**new results**
- We have made several theoretic contributions:
	- we proved that the optimal encoder $q_{\phi}$ trained to form representations of images that are coherent across noise levels (as specified by the mathematical form of the guidance score) smooths the latent space, addressing the “latent holes” phenomenon found in VAE representations (Appendix B1)
	- we also proved (using the specific geometry induced by the additive guidance) that the optimal representation is one that disentangles multi-scale features (Appendix B2)
	- additionally, we argue that our decoder is particularly well suited to data that contain multi-scale structure, using the analytic framework in Dai and Wipf (ICLR 2019), given by reviewer p7dT, Ref [4]. (Appendix B3)
- We have new numerical results on dSprites:
	- we trained our model on dSprites dataset, as requested by many reviewers. The results are in the Metrics section of the Appendix (Appx A7)
	- we measured disentanglement performance using two common metrics, Modularity and DCI Disentanglement. We chose these two metrics because, as we have discussed with reviewer p7dT, DCI is correlated strongly with other metrics aside from Modularity on the dSprites dataset (as shown in Locatello (2019)).
	- despite the lack of a rigorous hyperparameter search, we find that the metrics on our model are similar to the mean values produced by VAE-based models (in the Locatello 2019 paper).
	- this shows that even though our model does not have specific terms in the objective that encourage disentanglement aside from the $\beta$ parameter that controls the strength of the KL divergence.
	- this result is surprising given that the multiscale inductive biases of our model (that allow for near-SOTA disentanglement performance on CelebA) do not necessarily match the feature distribution in the dSprites dataset, which we measure empirically using our model

**changes to manuscript in response to individual feedback**
- reviewer 2F6a: we highlighted key results in the Appendix in the main text (results and discussion sections), and trained our model on the dSprites dataset.
- reviewer 12xU: in addition to the proof, we edited the main text to discuss in more depth a) how our model compares to other methods from Appendix A9, and b) the empirical metrics from Appendix A7.
- reviewer p7dT: in addition to the proof and new numerical results on dSprites, we emphasized the multi-scale aspect of our model in the main text and rephrased our claim in the abstract. We also updated the name of our model from DiVA to SAMI.
- reviewer qY7s: as requested, we measured the reduction in variance induced by the representation for the model trained on CelebA, and included this result in the main text. Further empirical results are in Appendix A6. We also referenced the most competitive model from Table 3 in the main text, as requested.

---

### Meta-Review · Area_Chair_7HMt · 2026-01-06

**Summary:**

This paper proposes a combination of diffusion models and VAEs for learning disentangled representations. Reviewers raised several concerns, including a lack of careful discussion of how the method differs from existing work, the experiments not being comprehensive enough. All original reviews recommended rejection. I believe 0-2 reviewers might have slightly increased their score, but even then, I think the paper should be rejected.

**Reviewer Concerns:**

The authors added experiments on dSprites and computed more disentanglement metrics. I suspect some reviewers might have found this sufficient to slightly increase their score, but I also suspect they might have found it to still be insufficient. Additionally, I don't think the concern about how the method differs was fully addressed.

**Reviewer Scores:**

I believe reviewer 12xU might have raised their score to a 4 and that reviewer qY7s might have raised their score to a 6.

---

### Decision · Program_Chairs · 2026-01-26

Reject